# Coding and noncoding landscape of extracellular RNA released by human glioma stem cells

Zhiyun Wei [1], Arsen O. Batagov[2], Sergio Schinelli[3], Jintu Wang[4], Yang Wang[1], Rachid El Fatimy[1], Rosalia Rabinovsky[1], Leonora Balaj[5], Clark C. Chen[6], Fred Hochberg[7,8], Bob Carter[7], Xandra O. Breakefield[5] & Anna M. Krichevsky[1]

Tumor-released RNA may mediate intercellular communication and serve as biomarkers. Here we develop a protocol enabling quantitative, minimally biased analysis of extracellular RNAs (exRNAs) associated with microvesicles, exosomes (collectively called EVs), and ribonucleoproteins (RNPs). The exRNA complexes isolated from patient-derived glioma stem-like cultures exhibit distinct compositions, with microvesicles most closely reflecting cellular transcriptome. exRNA is enriched in small ncRNAs, such as miRNAs in exosomes, and precisely processed tRNA and Y RNA fragments in EVs and exRNPs. EV-enclosed mRNAs are mostly fragmented, and UTRs enriched; nevertheless, some full-length mRNAs are present. Overall, there is less than one copy of non-rRNA per EV. Our results suggest that massive EV/exRNA uptake would be required to ensure functional impact of transferred RNA on brain recipient cells and predict the most impactful miRNAs in such conditions. This study also provides a catalog of diverse exRNAs useful for biomarker discovery and validates its feasibility on cerebrospinal fluid.

[1] Department of Neurology, Brigham and Women's Hospital and Harvard Medical School, HMS Initiative for RNA Medicine, Boston, MA 02115, USA. [2] Vishuo Biomedical, #3-33 Teletech Park, 20 Science Park Road, Singapore 117674, Singapore. [3] Department of Drug Sciences, University of Pavia, Pavia 27100, Italy. [4] Beijing Genomics Institute, Shenzhen 518083, China. [5] Department of Neurology and Radiology, Massachusetts General Hospital and Program in Neuroscience, Harvard Medical School, Charlestown, MA 02129, USA. [6] Neurosurgery Department, University of Minnesota, Minneapolis, MN 55455, USA. [7] Department of Neurosurgery, University of California, La Jolla, San Diego, CA 92093, USA. [8] Scintillon Institute, San Diego, CA 92121, USA. Correspondence and requests for materials should be addressed to A.M.K. (email: akrichevsky@bwh.harvard.edu)

ntercellular communication within complex biological sys-
tems, such as cancer and its host microenvironment, via
"horizontal" RNA transfer, is an expanding area of research[1].
Extracellular RNAs (exRNAs) are packaged into various extra-
cellular complexes, including microvesicles (MVs), exosomes, and
non-vesicular ribonucleoprotein complexes (RNPs)[2, 3]. MVs and
exosomes, broadly called extracellular vesicles (EVs), are released
and taken up by various cells, thereby transferring their content.
This process likely plays a role in cancer development and
manipulation of its microenvironment[4]. However, methodologies
are only beginning to emerge for characterizing the exRNA
landscape and monitoring levels of individual coding and reg-
ulatory exRNAs. exRNA mostly consists of small RNA species
(<200 nt); and the majority of reports to date focus on
miRNA[5, 6]. As a critical step toward understanding the biological
impact of exRNA release and transfer, we investigated the com-
plete spectrum of cancer-derived exRNAs, and the enrichment of
specific RNA classes and individual species. By creating cDNA
libraries of both small and long exRNA, and reducing the ligation
bias favoring miRNAs, we identified a diverse and highly distinct
composition of exRNA in MVs, exosomes, and RNP complexes.
Furthermore, semi-absolute quantification of RNAseq, performed
using RNA spike-in molecules, allowed us to monitor the levels of
various RNA classes and species in these extracellular complexes.

This work focused on glioblastoma (GBM), the most common
and aggressive brain tumor, as an important model for
investigation of cancer-derived exRNA. As proliferating and
invading GBM cells migrate through brain parenchyma, they
interact with the changing landscape of extra-tumoral stimuli and
actively modulate this landscape[4]. Communication between
tumor cells and diverse normal cells in the brain is nevertheless
one of the least investigated aspects of glioma biology. We
employed low-passage patient-derived tumorigenic GBM cell
cultures that represent the most therapy-resistant stem-like cell
population (GSC), and are considered the core cell type within
the tumor. Analysis of GSC cellular and extracellular RNA, along
with the transcriptome of primary human and mouse cells of the
brain microenvironment (neurons, astrocytes, endothelial cells,
and microglia) enables us to predict the most impactful miRNAs
and expand the repertoire of potentially transferred exRNAs far
beyond the classes of miRNAs and mRNAs. We also demonstrate
that MVs, large vesicles of 0.2–0.8 µm, most closely mirror the
cellular transcriptome and thus present a highly promising but
yet poorly explored source of liquid biopsy biomarkers.

## Results

**Sequential filtration-based exRNA isolation.** To characterize
exRNA released by patient-derived GBM cells in various com-
plexes, we assessed several technical approaches. EV and exRNA
isolation protocols can be generally categorized into three major
groups: based on ultracentrifugation (UC), precipitation using

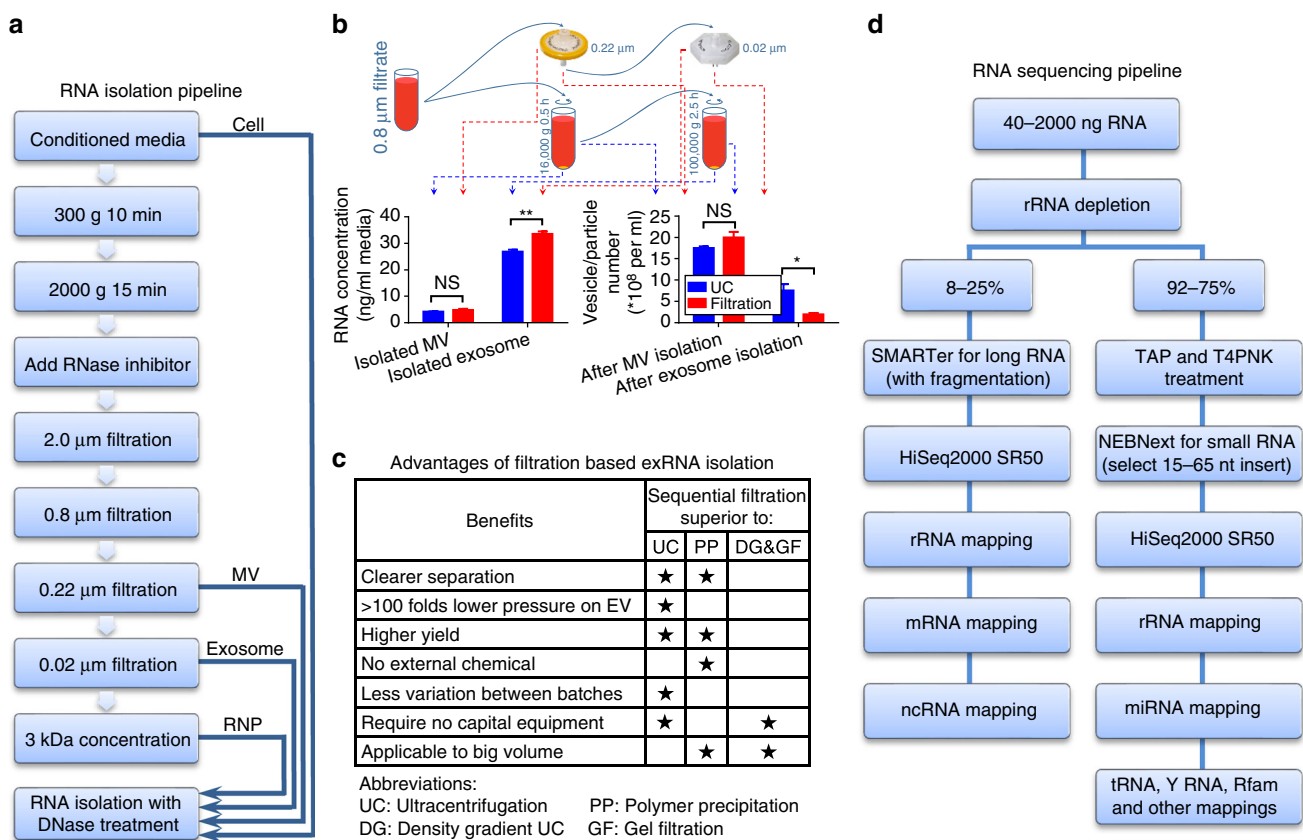

**Fig. 1** Flowchart of the exRNA fractionation and sequencing. **a** The pipeline of the filtration-based exRNA isolation. Following removal of cells and cellular debris by low speed centrifugation, the supernatants were filtered through a sequence of reduced pore sizes (2.0, 0.8, 0.22, and 0.02 µm) to separate the extracellular fractions, and a final concentrator with the cutoff of 3 kDa was applied to collect the remaining small particles. **b** The aliquots of conditioned media after 0.8 µm filtration were used for MV and exosome isolation, either by ultracentrifugation (UC) or filtration, and the RNA yield of these fractions compared. The number of remaining vesicles/particles was compared in UC supernatant and filter flow-through. $N = 4$ aliquots of conditioned media. All bars represent mean ± SEM. **c** Comparison between the filtration-based exRNA isolation and other common exRNA isolation methods. The stars mark superior characteristics of sequential filtration over other methods. **d** The optimized pipeline for the broad coverage, minimally biased RNA-sequencing. RNA of 15–65 nt was selected for the small RNA libraries, to reduce the overwhelming levels of tRNAs. NS, not significant; *$p < 0.05$; **$p < 0.01$; $t$-test

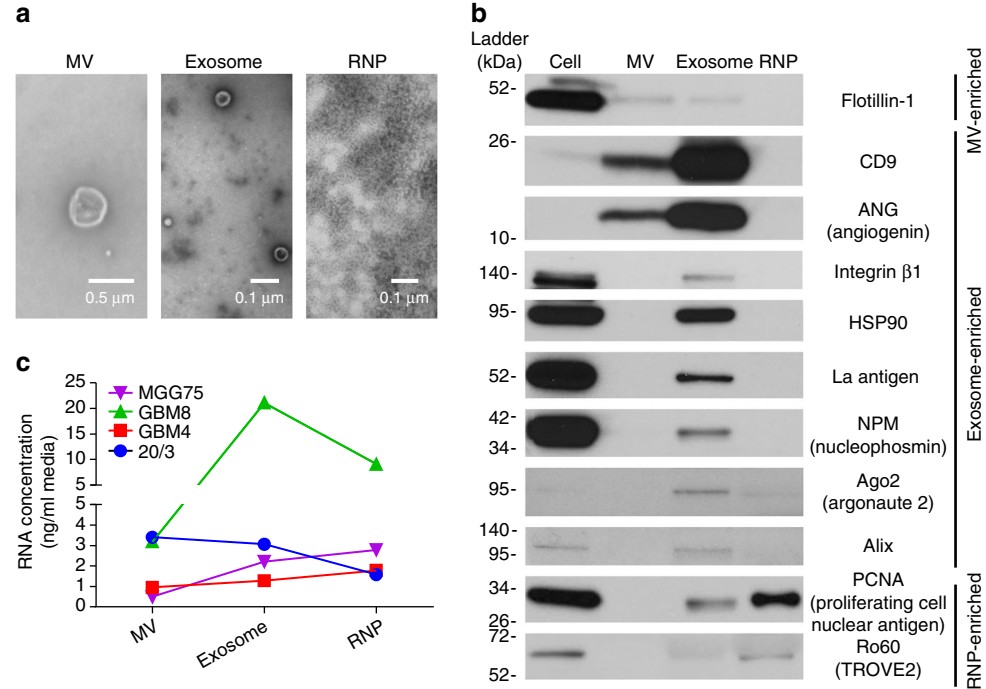

**Fig. 2** Quality control of the fraction separation. **a** Transmission electron microscopy of EVs and RNPs isolated using the sequential filtration protocol. TEMs were replicated three times. **b** Protein markers, identified by the western blotting, verified the separation of extracellular fractions. Equal protein input of 50 μg per lane was used. Western blots were replicated twice. **c** RNA yields of extracellular fractions from different GSC cultures

chemical polymers (PP), such as polyethylene glycol, and fractionation, including density gradient UC and gel filtration (DG&GF)[7]. Since specific markers or physical parameters for the various types of EVs and extracellular RNPs are still not clearly defined, UC remains the most commonly used approach to isolate the entire vesiculome[8]. However, based on nanoparticle tracking analysis (NTA; NanoSight) and fluorescent dye-binding quantification (RiboGreen), the yield of EVs and exRNA isolated by this technique is relatively low (20–40%) (Supplementary Fig. 1). Furthermore, this procedure yields a highly heterogeneous mix of EVs and RNP/LNP (liponucleoprotein) particles[9, 10]. To separate EVs and RNPs according to their physical size and improve the yield of exRNA isolation, we developed a sequential filtration (SF) protocol (Fig. 1a). This protocol offers several advantages over current methods, including low pressure on EVs, better separation between EVs and RNPs, higher RNA yield, and scalability (summarized in Fig. 1b, c and Supplementary Fig. 1). However, the extended hands-on time of the filtration procedure (Supplementary Fig. 2) and the separation solely on the basis of size are limitations of this method. Also, retrieval of EVs from the filtration membranes is inefficient and could potentially alter their structure; therefore, the utility of this protocol for functional EV analysis needs additional evaluation.

Established glioma cell lines have very limited capacity to reflect GBM biology[11]. We utilized previously characterized low-passage GSC cultures derived from four primary human heterogeneous GBM tumors for the exRNA profiling[12, 13]. These cells were grown as neurospheres in serum-free medium, to maintain their initial properties and transcriptional profiles and, therefore, better reflect tumor biology[14]. Transmission electron microscopy confirmed the presence of EVs/particles in the corresponding extracellular fractions isolated from GSC cultures (Fig. 2a). Distinct profiles of several protein markers exhibited by cellular and extracellular fractions served to confirm the purity of fractions and the lack of cellular contamination in the MV and other extracellular fractions (Fig. 2b). GSC cultures derived from different patients varied in the amount of exRNA released (Fig. 2c), ranging between 5.4 and 38.0 ng/ml accumulated in culture over 7 days. Considering that 3–10 μg of cellular RNA was isolated from 1 ml of the corresponding GSC neurosphere cultures (~1.3 million cells), between 0.05 and 0.7% of cellular RNA is accumulated in the extracellular space in 7 days. Of note, total exRNA yield varied ~7-fold among the GSC cultures, and the proportion of exRNA associated with different extracellular complexes also varied between the GSC types, suggesting the variations between cultures reflected intrinsic properties of different tumors. Analytical RNA profiles examined by the Agilent 2100 Bioanalyzer indicated high-quality cellular RNA (RIN > 9.5) with sharp rRNA peaks and no sign of degradation (Supplementary Fig. 3). In contrast, exRNA exhibited mostly short RNA profiles (below 200 nt) with intact rRNA peaks detectable in large MVs, but not exosomes and RNPs.

**Technical considerations for RNAseq.** The protocols commonly utilized for small RNA library construction are based on adaptor ligations to the 5′-phosphate and 3′-hydroxy ends of RNA, the modifications characteristic for miRNA, and thus favor miRNA[15]. In order to characterize RNA content in a minimally biased way, we utilized sequential treatments with tobacco acid pyrophosphatase (TAP) and T4 polynucleotide kinase (T4 PNK) to create more uniform 5′ and 3′ ends for various types of transcripts, leading to their more accurate representation in the cDNA libraries[16, 17]. The caveat is that this end-modifying procedure leads to an overwhelming abundance of rRNA reads in cellular and exRNA samples, and reduces the sequencing depth for other RNA classes. Therefore, we included an rRNA depletion step in the protocol that reduced rRNA reads remarkably (Fig. 3a).

Unlike the established strategies for normalization of cellular RNAseq data sets, which utilize total mapped reads as the normalization factor, there are no adequate standard for

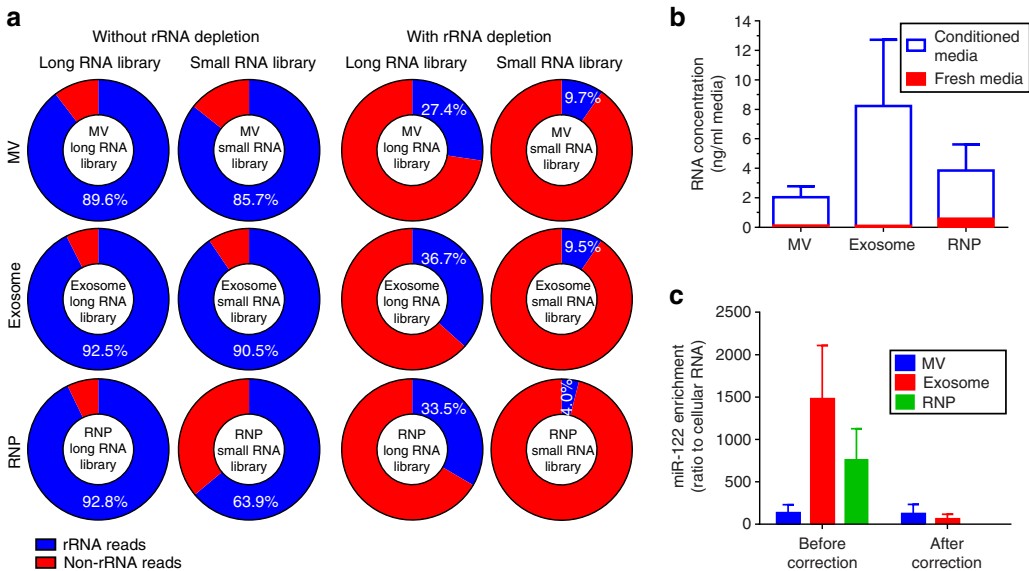

**Fig. 3** rRNA depletion and media correction are warranted for the deep RNAseq analysis. **a** Comparison of the exRNA libraries prepared with and without rRNA depletion. The percent of rRNA reads is shown, indicating that the majority of exRNA reads in non-depleted libraries represent rRNA. Sequencing depth of other exRNA classes is substantially increased by the rRNA depletion. **b** Quantification of total RNA in conditioned and fresh media indicates that the vast majority is cell derived (n = 4 GSC cultures). Nevertheless, quantification of specific RNA species can be skewed by the media, as illustrated in **c**. The levels of miR-122 were assessed in exRNA isolated from the fractions of conditioned and fresh media. miR-122 enrichment in exRNA (n = 4 GSC cultures) was calculated pre- and post-correction to the levels in fresh media, as described in Supplementary Fig. 6. miR-122 was highly enriched in the GSC exRNA, relative to its intracellular level, with up to 1500-fold enrichment in the exosomes before media correction. miR-122 enrichment in GSC exosomes became marginal upon correction. All bars represent mean ± SEM

comparative quantitative assessment of cellular vs extracellular RNAs. Since the proportion of total small RNA, and specific types of RNA is vastly different in cellular and extracellular RNA (Fig. 4a and Supplementary Fig. 3), normalization to total mapped reads is not optimal. As recently proposed[18], we utilized spike-in RNA for normalization, and quantified the abundance of RNA species as fmol per μg of total RNA. Further, since fresh culture media (FM) contain RNA that is co-isolated with cell-derived exRNA[19], we also assessed the interference of FM RNA with downstream analysis of GSC exRNA. With this goal, we isolated three RNA fractions from the corresponding FM, using the same filtration procedure, and subjected them to RNAseq in parallel with the GSC exRNA. Approximately 1 ng/ml total exRNA was isolated from FM, and it was largely associated with the RNP fraction (Supplementary Fig. 4). Overall, FM contributed 1.3–15% to the exRNA isolated from conditioned media, varying between the fractions (Fig. 3b). Although this amount of contaminating RNA is small, it can still affect the results of exRNA enrichment analysis. For example, miR-122 falsely showed exosomal enrichment (Fig. 3c), consistent with previous reports[20, 21], due to its abundance in B-27 supplement (Supplementary Fig. 5). Based on this observation, we included FM RNA data set in our RNAseq analysis pipeline, to provide the baseline for GSC-derived exRNA (Supplementary Fig. 6). The results described below were obtained using the media correction.

**EVs and RNPs exhibit distinct RNA composition.** To characterize the repertoire of GSC cellular and extracellular RNA, we sequenced the libraries of small and long RNAs, and first normalized the number of reads for each RNA class to the total number of non-rRNA reads within the library, thereby removing the confounding factor of variable rRNA depletion efficiencies. All reads generated on the long RNA libraries (that generally included transcripts longer than 100 nt), were classified as either

mRNA or non-coding RNA. Although in three out of four GSC cultures, cellular mRNA reads accounted for 20–35% of long RNA libraries, the mRNA proportion in EV fractions was below 10%, and even lower in RNPs (Fig. 4a). The reads obtained from small RNA libraries were first mapped to the most accurately annotated miRNA database (miRBase), and subsequently to other databases with equal mapping priority. In total, all annotated RNA species were categorized into 14 classes (Fig. 4a). Since small RNA libraries were built on 15–65 nt transcripts, the vast majority of the mapped reads represent fragments rather than full-length transcripts, with exception of miRNA and piRNA reads. Despite the remarkable heterogeneity of the GSC cultures, different extracellular fractions exhibited common characteristics of their RNA repertoires. Some of the most distinct features of GSC-derived exRNA are summarized as follows: (1) mRNA exons and snoRNAs are depleted, compared to cellular RNA, in all extracellular fractions; (2) all extracellular fractions, especially non-vesicular RNPs, are highly enriched in specific Y RNA fragments of largely unknown functions; (3) MVs and exosomes differ in their RNA composition, with mRNAs being relatively more enriched in MVs and miRNAs in exosomes; (4) RNP fractions have a highly distinctive RNA repertoire, with tRNA and Y RNA fragments strongly enriched, and snRNA and repeats reduced. The predominance of tRNA and Y RNA fragments in RNP is reflected in the corresponding sharp ~32 nt peak observed in the length distribution profile of reads (Supplementary Fig. 7). The relative abundance of piRNA and scRNA (small cytoplasmic RNA) fragments was also higher in RNP fractions. However, the most abundant individual piRNA and scRNA species were identical or highly homologous to major tRNA and Y RNA fragments, respectively (Supplementary Fig. 8). Whether such identical sequences indeed belong to two functionally distinct classes of transcripts, or tRNA and Y RNA are commonly mis-identified due to the poor quality of databases, is unknown. In addition, recently discovered circular RNAs have been reported as

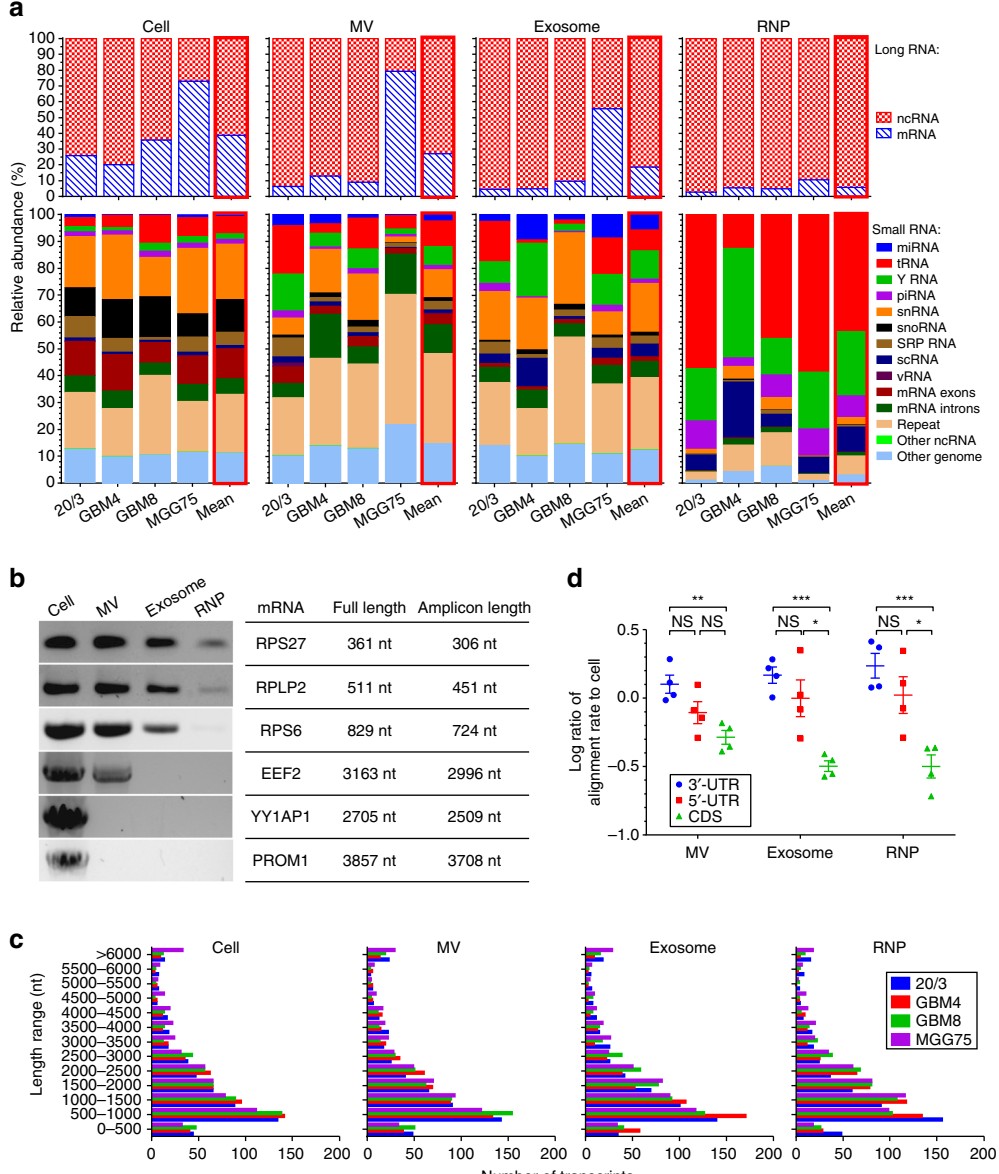

**Fig. 4** Relative composition of diverse RNA classes in cellular and extracellular compartments (MVs, exosomes, and RNPs) in different GSC cultures. **a** The top panels exhibit relative RNA composition in long RNA libraries, and the bottom panels depict the composition in small RNA libraries. The data were normalized to the total number of annotated non-rRNA reads. The bars framed in red represent the mean values of four GSC cultures.
**b** RT-PCR analysis (with equal input of total RNA) of selected mRNAs abundant in exRNA, demonstrates the presence of nearly full-length short, but not long messages in the extracellular fractions. Long RT-PCRs were replicated twice. **c** Long RNA libraries-based analysis of the length distribution of 500 most abundant mRNAs suggests no length preference for shorter parent transcripts in the extracellular fractions. **d** Analysis of the RNAseq reads mapped to mRNAs indicates that UTR regions were more enriched than CDS regions in the extracellular fractions ($n = 4$ GSC cultures). mRNA reads were aligned to the coding sequences (CDS), 5′-UTRs, and 3′-UTRs separately, and the alignment rates for each extracellular fraction were normalized to the corresponding cellular fraction. The log-transformed ratios of the alignment rates were compared among the three regions. Error bars represent mean ± SEM. NS, not significant; *, $p < 0.05$; **, $p < 0.01$; ***, $p < 0.001$; t-test

enriched in exRNA[22], and although our RNAseq protocol was not optimized for their quantification, several circular RNA species were detected (Supplementary Table 1).

**UTR fragments are released more than ORFs and intact mRNAs.** The length of intact cellular mRNAs varies between 350 and 12,000 nt, with the average around 2000 nt[23]. Considering the small size of EVs, the maximum length of an mRNA that could be packaged is still an open question. Previous high-throughput studies have not discriminated between full-length mRNAs and

fragments in exRNA[24, 25]. On average, our RNAseq detected 17,148 intracellular mRNAs in GSCs, whereas 17,219, 11,592, and 11,819 mRNAs were detected in MV, exosome, and RNP fractions, respectively. Many of the most abundant extracellular mRNAs corresponded to relatively short transcripts, including mRNAs for various ribosomal proteins (Supplementary Data 1). However, some abundant reads in EVs corresponded to long mRNA transcripts, such as PROM1 and EEF2. To examine whether intact mRNAs are present in extracellular fractions, we designed PCR primers for selected mRNAs to provide their full-length amplification. As shown in Fig. 4b, near-complete short

**a**

Evenness factors of RNA species abundance distribution

| RNA category | Cell | MV | Exosome | RNP |
|---|---|---|---|---|
| All long RNA | 6.37±1.49 | 4.71±1.53 | 4.67±1.04 | 4.92±0.47 |
| --mRNA | 11.70±0.89 | 12.12±0.36 | 14.27±3.85 | 16.30±2.45 |
| --ncRNA | 2.59±0.72 | 4.26±0.89 | 2.89±1.76 | 3.81±0.40 |
| All small RNA | 3.94±0.26 | 3.12±0.10* | 2.26±0.19** | 0.91±0.15*** |
| --miRNA | 6.09±0.34 | 5.27±0.12 | 5.39±0.35 | 4.76±0.30* |
| --snoRNA | 16.62±1.79 | 17.98±0.72 | 16.24±1.27 | 9.41±2.57 |
| --snRNA | 11.63±0.57 | 8.10±0.69** | 6.98±0.58** | 5.99±0.99** |
| --SRP RNA | 17.81±0.79 | 15.57±0.45* | 12.32±1.05** | 14.19±1.43 |
| --tRNA | 19.20±0.45 | 17.41±0.45* | 11.61±0.89*** | 8.48±0.45*** |
| --vRNA | 31.25±6.25 | 31.25±6.25 | 25.00±0.00 | 18.75±6.25 |
| --Y RNA | 11.27±0.98 | 8.33±0.56* | 6.54±0.27** | 5.88±0.38** |

**b**

Fold change of $X^2$ value to estimate heterogeneity difference

| RNA category | Cell | MV | Exosome | RNP |
|---|---|---|---|---|
| All long RNA | 1 | 1.340 | 1.720 | 1.660 |
| --mRNA | 1 | 0.967 | 2.746 | 2.905 |
| --ncRNA | 1 | 1.792 | 2.137 | 1.998 |
| All small RNA | 1 | 1.868 | 2.651 | 1.624 |
| --miRNA | 1 | 1.045 | 0.838 | 1.001 |
| --snoRNA | 1 | 2.034 | 1.762 | 2.196 |
| --snRNA | 1 | 2.060 | 1.995 | 4.266 |
| --SRP RNA | 1 | 1.073 | 3.520 | 3.069 |
| --tRNA | 1 | 0.423 | 0.916 | 1.889 |
| --vRNA | 1 | 0.692 | 0.068 | 0.958 |
| --Y RNA | 1 | 0.401 | 1.030 | 1.002 |

**c**

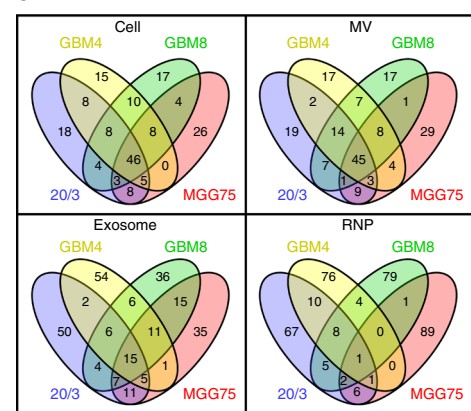

Fig. 5 Inequality and heterogeneity of the RNA repertoire of extracellular fractions. **a** Evenness factors, reflecting the inequality of abundance' distribution of the indicated RNA classes in various fractions. Higher evenness factors correspond to lower inequality. The classes of tRNA, Y RNA, snRNA, and SRP RNA, but not miRNA, contributed to the increased inequality of exRNA most significantly. Differential evenness factors of cellular and extracellular RNA suggest the selectivity of secretion ($n = 4$ GSC cultures). **b** For each RNA category, a sum of squared errors ($\chi^2$ value) was calculated among four GBM cultures, after normalization of each RNA species to the total number of reads in that RNA category. The $\chi^2$ value of each extracellular fraction was compared to the cellular fraction. Fold change of $\chi^2$ values higher than 1 reflects the increased heterogeneity. Heterogeneity either increased or decreased more than two-fold is highlighted in red and blue, respectively. **c** Venn diagrams depict the number of common species among 100 top abundant mRNAs, in four GBM cultures and their extracellular fractions, supporting the observation of higher heterogeneity of mRNA composition in exRNA than in cellular RNA. Significant differences between the cellular and extracellular fractions are depicted as following: *, $p < 0.05$; **, $p < 0.01$; ***, $p < 0.001$; $t$-test

mRNAs could be detected in cellular and all extracellular fractions, but detection of long mRNAs above 1000 nt was limited to cells and MVs only. These results suggest that either long mRNAs are excluded from packaging into exosomes and RNP complexes, or they are present in these fractions only as fragments. The latter appears to be the case, since the RNAseq demonstrates a similar representation of transcripts of various lengths in the intracellular and extracellular compartments (Fig. 4c). These data suggest that most exRNA reads corresponding to long mRNAs represent fragmented transcripts. Nevertheless, amplifying long RNA from low-input exRNA fractions is technically challenging and cannot be performed in a high-throughput manner, so we cannot exclude that some long full-length mRNAs are present extracellularly. Next, mRNA reads were aligned separately to the coding sequences (CDS), 5′-UTRs, and 3′-UTRs. The UTR regions, and especially 3′-UTRs, were significantly enriched in all extracellular fractions relative to CDS sequences (Fig. 4d), validating the previous observation[26] and suggesting differential release pathways for the protein-coding and regulatory sequences.

**Inequality of RNA representation increases in exRNA.** A relatively small number of the most abundant miRNA species

usually account for the majority of total miRNA in a given cellular context[27]. Consistent with this observation, 31 miRNA species accounted for 80% of the total miRNA in GBM8 cells, indicative of a diverse range of miRNA expression (or "inequality" of miRNA levels). Even fewer, 19 miRNA species, accounted for 80% of the total miRNome in GBM8 exosomes, suggesting a higher inequality of miRNAs in exRNA. Such comparison, however, relies on a randomly selected cutoff (80% used above). To compare the inequality of intracellular and extracellular transcripts more objectively, we developed two alternative strategies. The first one is an improved version of traditional evaluation named the evenness factor ($\varepsilon$), which defines that $\varepsilon$% of RNA species can account for (100−$\varepsilon$)% of total abundance. The second strategy is based on the Gini coefficient, commonly used for evaluation of inequality in economics. Of note, although both are objective and use no preset cutoff, higher $\varepsilon$ and lower Gini coefficient correspond to a more equal representation (or lower inequality). The detailed illustration of the concepts behind these evaluation systems can be found in Supplementary Fig. 9. Taking advantage of these parameters, we compared transcript inequality in cellular and extracellular fractions, either for the whole RNA library or a specific RNA class. The inequality of RNA levels was similar for cellular and extracellular long RNA libraries, but

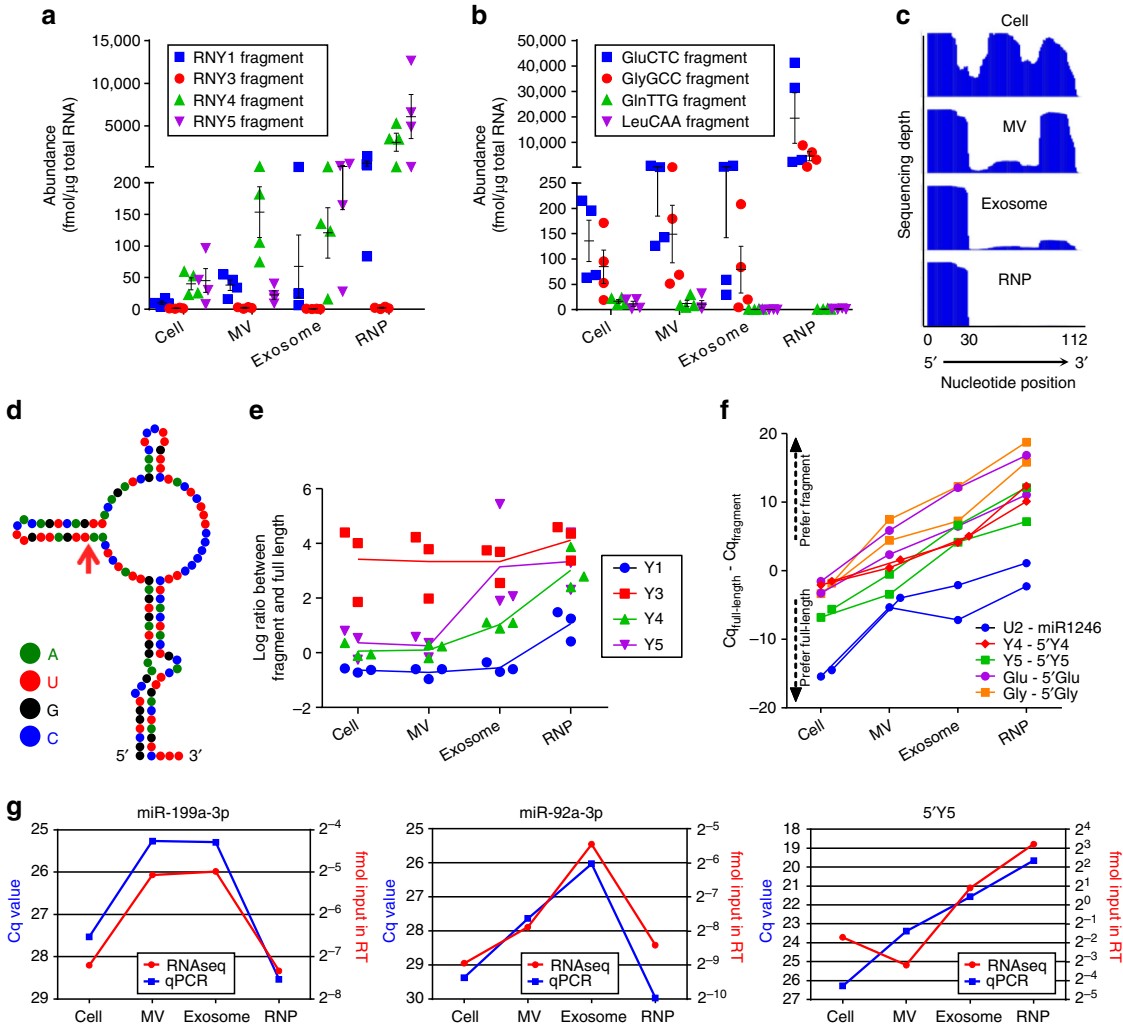

**Fig. 6** Y RNA and tRNA fragments are abundant and enriched in exRNA. **a**, **b** Specific Y RNA and tRNA species are among the most abundant in small RNA libraries ($n = 4$ GSC cultures). The reads corresponding to Y1, Y4, and Y5 are highly enriched in extracellular fractions, especially in RNP (**a**). The reads corresponding to specific tRNAs, such as GluCTC and GlyGCC, are highly enriched in exRNA, while others (e.g. GlnTTG and LeuCAA) are not (**b**). **c** Mapping coverage of Y1 reads indicates that the Y1 is precisely processed, and mostly its 5′ fragment is present in exosomes and RNPs, as evidenced by the steep peaks corresponding to the 5′-end 30 nt. These profiles are distinct from the more uniform full-length coverage observed for the cellular RNA, and to a lesser extent MV RNA. Similar analysis for other Y RNAs and tRNAs is presented in Supplementary Fig. 12. **d** The predicted secondary structure of Y1 RNA[30], Copyright (1993) National Academy of Sciences, USA, and the position of its cleavage (indicated by the arrow) that produces the 5′ fragment which is highly abundant in exRNA. **e** Quantification of Y RNA reads in long and small RNA libraries, demonstrates different fragment to full-length ratios in the cell and exRNA fractions ($n = 3$ GSC cultures; MGG75 was excluded due to very low abundance of the full-length Y RNA). The ratios are increased in extracellular fractions. **f** qRT-PCR analysis with primers specific to either full-lengths or fragments of several RNA species validates the enrichment of 5′ tRNA and Y RNA fragments in extracellular fractions. For each specific transcript examined, two lines represent GBM8 and 20/3 cells, respectively. **g** qRT-PCR analysis of selected transcripts confirms the quantitative character of the RNAseq pipeline. The blue dots represent qRT-PCR Cq values, while the red dots represent the results of RNAseq quantification in fmol. Both analyses were performed on the same set of RNA samples. Error bars represent mean ± SEM

increased significantly in small exRNA libraries (Fig. 5a and Supplementary Table 2). These results provide support for the idea of selective RNA incorporation into different exRNA fractions, that is still highly debated in the field.

**Heterogeneity of RNA repertoire increases in exRNA fractions.** Despite the genetic diversity of GSCs, cellular RNA class composition was similar among the four different GSC cultures analyzed, with the exception of MGG75 cells that expressed more mRNA than long ncRNA (Fig. 4a). However, the RNA repertoire of extracellular fractions was more heterogeneous among GSC cultures than that of cells. To estimate the heterogeneity of

cellular and extracellular RNA species across the GSC cultures, we first normalized the reads to the total read number within an individual RNA category, and then evaluated the sum of squared errors ($\chi^2$ value). The higher $\chi^2$ value reflects the higher diversity/ heterogeneity of GSC cultures in terms of RNA composition. As shown in Fig. 5b, relative to cellular RNA, the extracellular fractions were, overall, more heterogeneous in their composition in both long and small RNA libraries, as well as for the majority of specific RNA classes. This phenomenon was not caused by technical irreproducibility of the exRNA analysis, because independently analyzed exosomes produced by different passages of GBM8 cells were much more concordant than the pairwise-compared exosomes released by GSC cultures established from

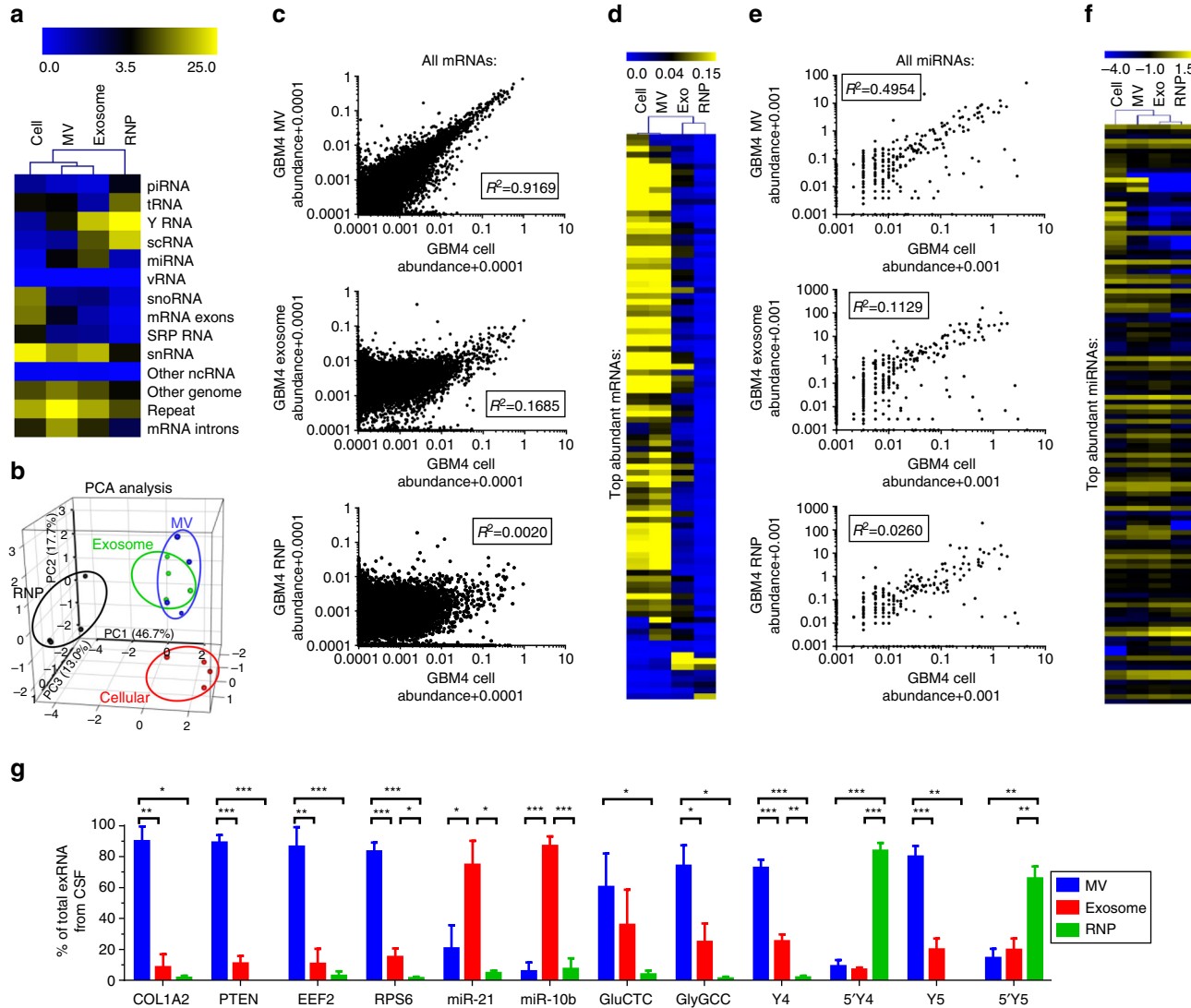

**Fig. 7** RNA repertoire of MV most closely reflects cellular RNA composition. **a** Heat map cluster analysis of RNA classes indicates relative similarity of the composition of MVs and the source cells. The scale bar represents the percentage of non-rRNA annotated reads. **b** PCA analysis of RNA classes. Different fractions are marked in different colors. Within each fraction, four dots represent four GSC cultures. **c**, **d** GBM4 MV represents the cellular mRNA content closely, and much better than exosomes or RNPs, based on the correlation analysis of all mRNA species (**c**), and cluster analysis of the top abundant mRNAs (**d**). The scale bar represents mRNA abundance. **e**, **f** Extracellular miRNA composition, in general, is less reflective of the cellular miRNome; nevertheless, MV fraction still remains the best simulator, based on the correlation analysis of all miRNA species (**e**), and cluster analysis of the top abundant miRNAs (**f**). The scale bar represents the log-transformed miRNA abundance. Similar analyses of other cell cultures can be found in Supplementary Figs. 14–17. **g** qRT-PCR analysis of exRNA fractions isolated from the CSF of GBM patients, using the same filtration-based procedure, indicates its applicability to clinical samples. The data verify preferential association of selected RNA species with different exRNA fractions in the human biofluid ($n = 4$ CSF samples). All bars represent mean ± SEM. *, $p < 0.05$; **, $p < 0.01$; ***, $p < 0.001$; t-test

different patients (Supplementary Fig. 10). Consistently, Venn diagram analysis showed less commonalities (or increased heterogeneity) for mRNAs in exosomes and RNPs, relative to cellular RNA (Fig. 5c). The heterogeneity of miRNA class, however, was similar between GSC cellular and exRNA compartments (Supplementary Fig. 11). Combined analyses of inequality and heterogeneity suggest that GSC cultures may utilize various sorting mechanisms for exRNA release.

**Y RNA and tRNA fragments are abundant and enriched in exRNA**. The reads corresponding to tRNA and Y RNA species constitute a significant proportion of the rRNA-depleted small exRNA libraries (8.5% in cells, 13.5% in EVs, and 67.5% in RNPs, Fig. 4a). All four human Y RNA (Y1, Y3, Y4, and Y5) and some

tRNA species are highly abundant in extracellular fractions, especially the RNP, reaching the quantities of up to dozens of pmol per µg of total RNA (Fig. 6a, b). Relative to let-7b-5p, one of the most abundant miRNAs in exRNA, those molecules are at least hundred times more abundant. Since human Y RNAs are 84–113 nt and tRNAs are 68–176 nt, and our libraries were constructed from 15 to 65 nt RNA, we reasoned that the reads represent fragments rather than full-length transcripts. Notably, the coverage analysis of the reads mapped to Y RNAs and tRNAs suggested the presence of specifically processed fragments (Fig. 6c and Supplementary Fig. 12), with the processing sites located within the loop domains that are known to bind several proteins[28–30] (Fig. 6d and Supplementary Fig. 12). Further integration of the small and long RNA data sets revealed that the ratios of fragment to full-length Y RNAs differed significantly among

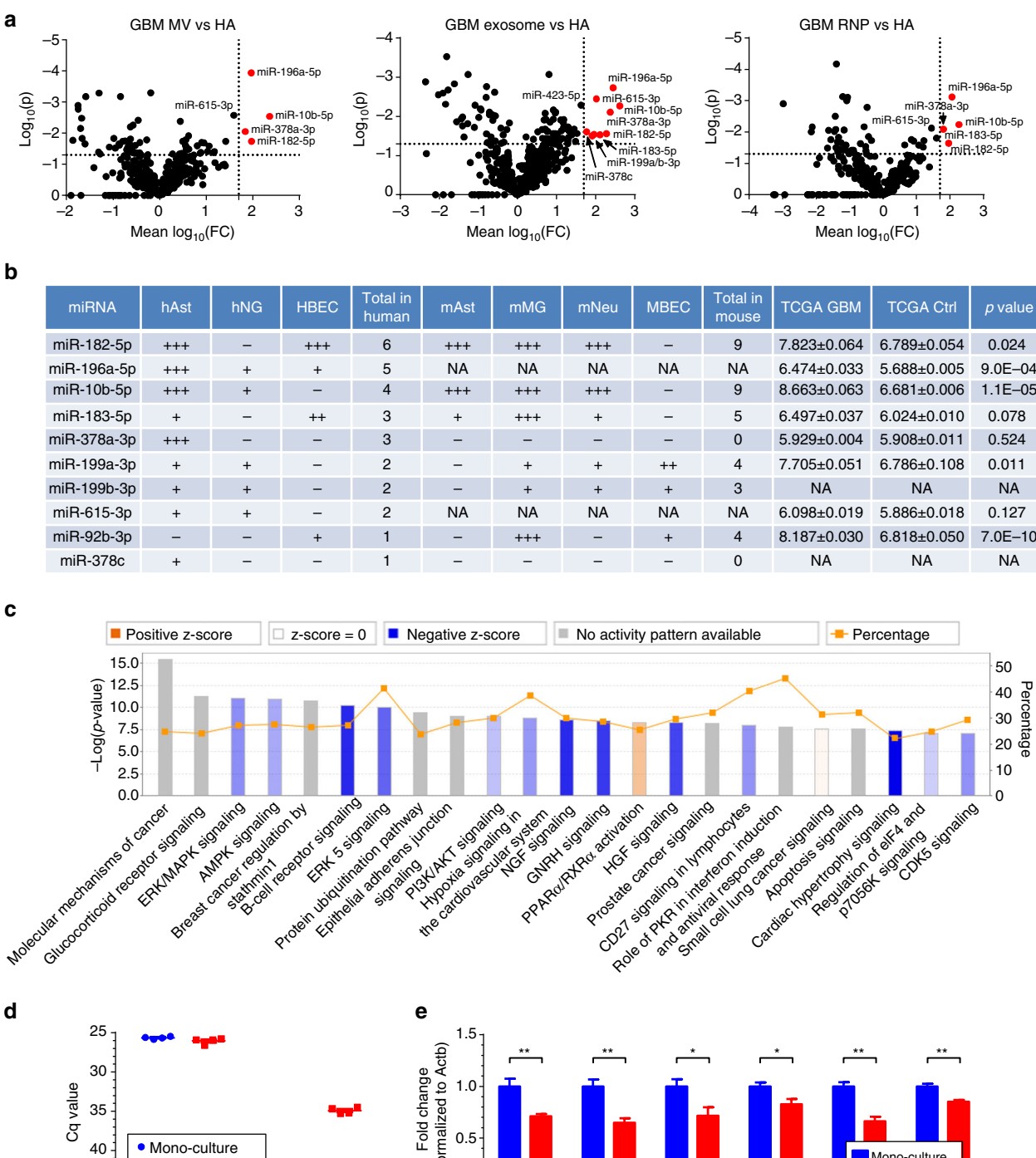

**Fig. 8** Comparative analysis of GSC-secreted miRNAs and cellular miRNome of the normal cells of the brain predicts the most impactful GBM miRNAs in tumor-to-microenvironment communication. **a** Extracellular fractions of GSC cultures were compared to primary human astrocytes (HA), based on the corresponding RNAseq data sets. The fold-changes in miRNA levels were log transformed and a *t*-test was applied to examine the significance of difference. MiRNAs with log-fold changes higher than 1.7, which corresponded to 50 times higher levels in the GSC-derived exRNA fractions relative to the recipient cells, and *p* < 0.05 (*t*-test), were defined as potentially impactful (colored in red). The horizontal axis of Volcano plots shows the log-fold difference, and the vertical axis shows the statistical significance. Similar analyses of primary human neuroglial and endothelial cells are shown in Supplementary Fig. 18. **b** A full list of most impactful GSC miRNAs for human and mouse astrocytes, neurons, microglia, or brain endothelial cells. The number of "+" symbols reflects the number of extracellular fractions in which an miRNA meets the indicated criteria as in **a**. Most of these miRNAs are also upregulated in the GBM tumors compared to non-neoplastic brain tissues in the TCGA microarray data set, as indicated in the three right columns (*n* = 496 GBM vs 10 control). **c** Top enriched IPA pathways for the validated mRNA targets of the most impactful miRNAs. Predicted activation and inhibition of pathways are labeled as orange and blue bars, respectively. The yellow line shows the percent of genes in each pathway that are validated targets. **d**, **e** Co-cultured with GBM8 neurospheres, primary miR-21-null astrocytes exhibit steady miR-21 levels (**d**) and downregulation of validated miR-21 targets (**e**). Cq value of miR-21 in mono-cultures was defined as 45 (undetectable expression). *N* = 4 wells in 24-well plate. All bars represent mean ± SEM. *, *p* < 0.05; **, *p* < 0.01; *t*-test

the Y RNA species (Fig. 6e). Overall, the fragment to full-length ratio was significantly higher in extracellular vs cellular fractions for all four Y RNA species (Fig. 6e), indicating that whereas full-length Y RNAs are highly abundant in the cells and MVs, specific fragments of them are enriched in the exosomes and RNPs. These data were further confirmed by qRT-PCRs (Fig. 6f). Moreover, such preference for fragments in the exosomes and RNPs was not limited to Y RNA. For example, full-length U2 snRNA (RNU2-1), which also serves as a precursor for miR-1246[31], and specific tRNAs are also released to a lesser extent in extracellular fractions than their corresponding processed products (Fig. 6f). As indicated by the experiment with unfractionated exRNA, fragmentation is a feature of exRNA rather than an artifact of the filtration procedure (Supplementary Fig. 13). Of note, qRT-PCRs quantification of the selected transcripts correlated well with the RNAseq results (Fig. 6g), supporting the accuracy of our RNAseq analysis.

**MVs most closely reflect cellular RNA composition**. Cancer-derived exRNA may serve as clinical biomarkers for disease diagnostics, prognostics and monitoring. An extracellular fraction that closely mirrors the deranged cellular transcriptome would be the most valuable for such applications. To date, mostly unfractionated, precipitated or 100,000 g-pelleted EVs have been explored as a source for potential biomarkers. To examine what type of extracellular complexes (MVs, exosomes, or RNPs) might serve as the closest proxy of the GSC transcriptome, we performed correlation and clustering analysis of their RNA composition. Overall, clustering analysis based on small RNA libraries demonstrated the highest similarity between the cellular transcriptome and MV content, and to a lesser extent to exosome content, while the RNP fraction had a highly distinct RNA composition (shown for GBM4 in Fig. 7a). This was further confirmed by principal component analysis (PCA) of all four GSC cultures (Fig. 7b). Similar analyses of mRNA and miRNA classes, and the most abundant transcripts within them, further supported this conclusion (Fig. 7c–f and Supplementary Figs. 14–17). Consistent with this idea, there were fewer miRNA species, and overall fewer different RNA species significantly enriched in MVs than in exosomes and RNPs (Supplementary Data 2). The observation that MVs provide a more accurate peripheral readout of the source cell content is in line with recent extracellular proteomic analysis[32]. Therefore, MVs which include large vesicles (200–800 nm) appear to be a good source for RNA biomarker discovery, and have the potential to outperform the more studied smaller exosomes.

**Analysis of exRNA complexes in cerebrospinal fluid**. To investigate the differential potential of exRNA fractions for biomarker discovery, we tested our sequential filtration method on cerebrospinal fluid (CSF) samples obtained from four GBM patients. As shown in Fig. 7g, sufficient amounts of total RNA for qRT-PCR analysis were isolated from each CSF fraction (MVs, exosomes, and RNPs), and distinct profiles observed for different RNA classes. Although we could not examine the resemblance of CSF exRNA profiles to those of parental tumor and neural cells due to unavailability of tumor material from the corresponding patients, the data indicated that, indeed, MVs and exosomes should be analyzed separately in biomarker discovery studies. Similarly to the results obtained on cultured GSC-derived exRNA, the CSF mRNAs were preferentially associated with MVs and miRNAs with exosomes, suggesting that MV enrichment is warranted for mRNA biomarkers, while exosomes represent a superior source for miRNA biomarkers. Of note, several mRNAs frequently mutated in GBM, including PTEN and COL1A2, were

detected in CSF MVs. The preference for full-length Y RNAs in CSF MVs, and their fragments in RNPs was also consistent with the observations made on GSC-derived exRNA.

**Exosomal miRNAs with high functional potential upon transfer**. A common, but still highly debated, hypothesis is that EVs and EV-associated exRNA are taken up by recipient cells and mediate intercellular communication. MiRNA transferred from GBM to cells in their microenvironment could have significant impact on the transcriptome of the peritumoral cells. We reasoned that the greatest impact would be mediated by those miRNAs that are most abundant in EVs/exRNA, while not or minimally expressed in recipient cells, such that their transfer would significantly alter their levels in recipient cells and thus affect the regulation of mRNA targets. To identify such miRNAs, we compared relative levels of all miRNAs in exRNA fractions from GSCs with those in the major types of human and mouse brain cells, including astrocytes, mature neurons, microglia, and brain-derived endothelial cells. Comparison of three GSC extracellular fractions with primary astrocytes is shown in Fig. 8a, and similar analysis for other cell types is presented in Supplementary Fig. 18. Summary of this analysis, performed on three extracellular fractions and four types of recipient cells, revealed a list of the GBM-derived miRNAs that might potentially have the strongest impact on the normal cells of brain microenvironment (Fig. 8b). In agreement with these data, most of the listed miRNAs were also found to be elevated in GBMs relative to the control brain tissues in The Cancer Genome Atlas (TCGA) data set (Fig. 8b). To predict the downstream effects of the 10 most impactful miRNAs on the recipient cells, we analyzed their direct mRNA targets previously validated by at least three supporting CLIP-Seq data sets in the starBase database[33]. In total, 2267 mRNA species interact with at least one of the impactful miRNAs. Based on the ingenuity pathway analysis (IPA), these targets are significantly enriched in many canonical cancer-related pathways and bioterms, including the molecular mechanism of cancer, ERK/MAPK signaling, PI3K/AKT signaling, and NGF signaling (Fig. 8c).

**Less than one copy per EV on average for most RNA species**. A substantial amount of transferred exRNA might be required to exert a functional effect in a recipient cell; however, the quantitative data for the levels of specific RNA classes and individual transcripts in exRNAs is very limited thus far. To address this issue, we performed a stoichiometric analysis of the EV-associated RNA. On average, GSC EVs (MVs and exosomes collectively) provided ~8.9 ng total RNA per ml of conditioned media. Considering the concentration $2 \times 10^9$ EVs per ml, measured by NTA, one EV contained ~4.45 ag total RNA, or ~0.445 ag non-rRNA, which corresponded to ~836 ribonucleotides. Similar analysis was performed for individual RNA classes and species. As shown in Table 1, rRNA and snRNA are present in more than one copy per EV on average. RNA classes represented by approximately one copy per EV include Y RNA and Y RNA fragments, tRNA fragments, lncRNA, and miRNA. One copy of mRNA or mRNA fragment can be found in ~10 EVs. Of note, quantification of individual RNA species revealed that only several specific molecules are present at the level of one copy per EV (e.g., RNY1, Y5 fragments, GluCTC tRNA fragment, U1 and U2 snRNA fragments). The most abundant miRNA species (e.g., let-7b, miR-21) were present at the level of one copy per 10 EVs approximately, consistent with the previous report[34]. The most abundant individual mRNA species were present at one copy per 1000 EVs approximately (e.g., TMSB10), and the levels of mRNAs most commonly mutated in GBM (e.g., EGFR, IDH1,

TP53, PTEN, and COL1A2) were not higher than one copy per 100,000 EVs approximately. Based on these data, continuous barrage, massive and/or highly selective uptake of EVs might be required to affect signaling or alter the phenotype of the recipient cells via their exRNA content.

To examine the functionality of transferred GSC-derived miRNAs, we established 3D transwell co-cultures of GBM8 neurospheres with primary mouse astrocytes, a system more physiological than the commonly used exposure of recipient cultures to super-concentrated EVs isolated from donor cell conditioned medium. In line with the low copy number of individual miRNAs per EV, we were unable to detect significant elevation of either miR-21 or miR-10b (top GBM-promoting miRNAs) in astrocytes co-cultured with GBM8. However, miR-21 became readily detectable in primary astrocytes established from miR-21 knockout mice upon their co-culture with GBM8 (Fig. 8d). Furthermore, this latter miR-21 transfer led to functional effects in these cells, i.e., repression of its previously validated mRNA targets (Fig. 8e)[35]. These data support intercellular transfer of miRNA via exRNA and suggest that, despite the apparently low levels in EVs, miRNAs may exert regulatory functions in recipient cells, albeit with specific conditions and highly sensitive methodologies required for detection.

## Discussion

exRNA studies have expanded rapidly and multidirectionally in this decade. Revealing the composition of exRNA complexes released by defined cell types remains one of the most fundamental milestones toward understanding the role of exRNA in intercellular communication, as well as discovery of RNA biomarkers for disease. This study provides the first minimally biased quantitative analysis of the exRNA released by tumor-derived cells from GBM patients. Despite the high heterogeneity of GSC cultures established from different tumors, and even higher heterogeneity of the exRNA they release, extracellular complexes share key characteristics. The RNA profiles of MVs, exosomes, and RNPs are highly distinct. They all display selectivity in their RNA loading compared with the cells, with the MVs being the most like the cells and the RNPs the least. Small and fragmented RNAs account for the majority of exRNA in all three types of complexes. Between 64 and 93% of all exRNA consists of fragmented rRNA (Fig. 3a), although there is little-to-no intact rRNA (Supplementary Fig. 3). miRNA, the most studied class of exRNA, constitutes <10% of non-rRNA exRNA (Fig. 4a). Consistent with some previous publications[36], and in contrast to others[20, 37–41], we found that miRNA species are relatively enriched in exosomes, but not in MVs or RNPs. Of note, the isolation approaches, the library preparation strategies, and the normalization methods utilized in these different studies were diverse, making the results not directly comparable. Other non-coding RNA species are more abundant, and some of them are enriched in EVs or RNPs. Notable among them are precisely processed tRNA and Y RNA fragments, associated with both EVs and extravesicular RNPs, and supported by observations of other cultured cells[20, 37] and body fluids[42].

The most common extracellular 30–32 nt-long 5′-tRNA fragments (5′-tRFs), also called 5′-tRNA halves or tiRNAs, are evolutionarily conserved molecules[43] produced by angiogenin (ANG), a member of ribonuclease A family[44]. This multifunctional ribonuclease regulates angiogenesis, cell proliferation and viability of cancer cells, as well as neuronal survival and stress response[43]. Specific 5′-tRFs are known to perform crucial functions, often associated with regulation of gene expression in stress response. They repress protein translation by displacing eukaryotic translation initiation factors eIF4E and eIF4G[45]; and modulate stress response by inducing formation of stress granules—cytoplasmic foci where untranslated mRNAs are transiently stored[44]. Importantly, specific 5′-tRFs (e.g., 5′Ala,

| Table 1 Quantification of selected RNA classes and individual species in copy number per EV | | |
|---|---|---|
| **Copy level** | **RNA class or individual species** | |
| More than 1 copy per EV | RNA class | rRNA fragment[a] |
| | | Repeat fragment |
| | | snRNA fragment |
| One copy per 1 EV approximately | RNA class | tRNA fragment |
| | | Intron fragment |
| | | Y RNA fragment |
| | | SRP RNA fragment |
| | | miRNA |
| | | Exon fragment |
| | | lncRNA |
| | | Y RNA |
| | | snoRNA fragment |
| | Individual RNA species | U2 snRNA fragment |
| | | U1 snRNA fragment |
| | | GluCTC tRNA fragment |
| | | RNY1 |
| | | RNY5 fragment |
| One copy per 10 EVs approximately | RNA class | mRNA |
| | | vRNA |
| | | vRNA fragment |
| | Individual RNA species | RNY4 fragment |
| | | RNY4 |
| | | RNY1 fragment |
| | | let-7b-5p |
| | | GlyCCC tRNA fragment |
| | | miR-21-5p |
| | | vRNA1-2 |
| | | vRNA1-2 fragment |
| One copy per 100 EVs approximately | Individual RNA species | U3 snoRNA fragment |
| | | miR-1246 |
| | | miR-10b-5p |
| | | RNY5 |
| | | vRNA1-1 fragment |
| | | vRNA1-3 fragment |
| | | vRNA1-3 |
| One copy per 1000 EVs approximately | Individual RNA species | RNY3 fragment |
| | | TMSB10 mRNA (top 1 mRNA) |
| | | miR-93-5p |
| One copy per 10,000 EVs approximately | Individual RNA species | ACTB mRNA |
| | | GAPDH mRNA |
| | | miR-132-3p |
| One copy per 100,000 EVs approximately | Individual RNA species | COL1A2 mRNA |
| | | RNY3 |
| | | miR-34a-5p |
| | | IDH1 mRNA |
| | | EGFR mRNA |
| 1 copy per 1,000,000 EVs approximately | Individual RNA species | TP53 mRNA |
| | | PTEN mRNA |

[a]The transcripts were defined as fragments if present and quantified in small RNA libraries

5′His, and 5′Cys) selectively regulate translation of subsets of mRNAs, both capped and uncapped, and therefore reprogram protein synthesis[45]. They may also function in a miRNA-like manner[46]. ANG is upregulated in GBM[47, 48] and is one of the proteins most highly secreted by glioma cells[49]. Despite its high abundance, functions of ANG, including its extracellular activity and its exRNA products remain to be investigated. Remarkable enrichment of both ANG and 5′-tRFs observed in GSC-derived exosomes (Figs. 2b and 6f) suggests that tRNA cleavage may occur in exosomes, outside of the cells.

Another highly abundant but poorly studied classes of non-coding exRNAs are Y RNAs and their specific 5′ fragments. The majority of information about Y RNA has come from studies of bacteria and invertebrates. In vertebrates, Y RNAs are expressed in all tissues and cells and have been proposed to participate in many important cellular functions[50, 51]. Their lower stem domain, which recruits chaperone Ro60 and exoribonuclease PNPase, may play a role in RNA quality control and degradation of misfolded RNAs[50]. The upper stem domain may participate in initiation of chromosomal DNA replication[52]. With biogenesis and activity independent of Dicer and Ago2[53], Y RNA fragments do not appear to silence gene expression in a miRNA-like manner[54]. Recent studies reported that Y RNA fragments may be involved in histone mRNA processing[55] and cell damage[56]. Interestingly, Y5 fragment, the most abundant in exRNA, is proposed to specialize in surveillance of ribosomal RNA[57]. Studying the functions of these precisely processed, highly enriched extracellular transcripts represents an exciting new avenue in RNA biology.

In addition to non-coding RNA, we detected low levels of mRNA reads with UTRs relatively enriched compared to the coding regions in exRNA. The mechanism underlying this enrichment and its biological impact has yet to be investigated. Methylation and other RNA modifications of UTRs[58, 59] might cause differential sorting or stability, leading to enrichment in exRNAs. UTRs and their fragments might function as molecular sponges for various regulatory molecules, including miRNA, translation factors (e.g., eIF4F and ribosomal complexes), other RNA-binding proteins, and thereby exert functions in recipient cells. Altogether, our data indicate the strong preference of processed fragments for multiple classes of RNA, protein-coding and non-coding, in exRNA (Fig. 6). The co-packaging of processed RNA with various RNA-binding proteins, including RNases (e.g., ANG) and effecter complexes (e.g., Ago2), and the significant depletion of full-length RNA in exosomes, suggests that they may function as an exRNA processing machinery with specialized autonomous functions.

Although several studies have explored the RNA bound to specific secreted or circulating proteins, such as high-density lipoprotein (HDL)[3] and Ago2[60], overall little attention has been paid thus far to the non-vesicular exRNA complexes that account for the significant proportion of exRNA (Fig. 2c). We present data indicating that extracellular non-vesicular RNPs exhibit a RNA signature readily distinguishable from EVs. Notably, ncRNA accounts for nearly entire RNA population in RNPs, with a large prevalence of tRNA, Y RNA and their products, and depletion of miRNA species (Figs. 4 and 6), consistent with the analysis of breast cancer extracellular RNPs[20]. Although extracellular RNPs might be highly heterogeneous, this collective data suggest the key pathways involved in the biogenesis. Of note, we found that massive levels of albumin, an abundant component in B-27 supplement, hamper the protein characterization of extracellular RNPs, and markers of RNPs have yet to be defined. Benefiting from an efficient (>95%) albumin-depletion protocol, we were able to compare protein content in MVs, exosomes, and non-vesicular RNPs using immunoblots. Interestingly, two nuclear proteins PCNA and Ro60 appeared enriched in RNPs but not in EVs. While the co-localization of Ro60 with its binding Y RNA partners in RNPs may point to the functionality of this extracellular complex, the role of extracellular PCNA, a protein involved in DNA replication, remains to be elucidated. Although the possibility of direct non-vesicular, RNP-mediated RNA uptake and function in recipient cells is currently unexplored, it is supported by the utility of RNPs for RNA and drug delivery[61, 62]. Regardless of the potential biological function of RNPs, their associated RNAs expand the repertoire of potential biomarkers that should be further explored in body fluids, in parallel to EV transcripts.

The key outcome of our work, overall, is an expansion of the repertoire of exRNAs released by GSCs in different vehicles, with functional and biomarker potential, far beyond the class of miRNA. This conclusion challenges the commonly assumed predominant role of miRNA in exRNA-mediated intercellular communication. It further points to the need for in-depth investigation of other classes of exRNAs and their impact on the physiology of recipient cells and use as biomarkers. The future should bring the development of novel experimental techniques and computational resources for integrating complex expression data sets into comprehensive biologic networks and biomarker discovery.

## Methods

**GBM stem cell cultures.** Human low-passage (below 20) GBM cells (kind gift from Dr. Hiroaki Wakamoto, MGH) were cultured as neurospheres in Neurobasal medium (Gibco) supplemented with 3 mM GlutaMAX (Gibco), 1× B-27 supplement (Gibco), 0.5× N-2 (Gibco), 20 ng/ml EGF (R&D systems, MN), 20 ng/ml FGF (PEPROTECH, NJ) and 0.5% Antibiotic-Antimycotic Solution (Corning), and passaged by NeuroCult Chemical Dissociation Kit (Mouse) (Stemcell Technologies, Canada) following the manual. Approximately $5 \times 10^5$ dissociated cells were seeded per 10 cm dish (Corning) in 10 ml fresh media, and 1/3 volume of fresh medium was added every 3 days. Mature neurospheres, typically formed in 7–10 days, were dissociated and replated. All cells were tested for mycoplasma. Human cells were used in accordance with the policies of institutional review boards at Brigham and Women's Hospital.

**Primary cultures of normal brain cells.** Brain cortices of E18 and P1 C57BL/6 mice were dissected for primary cultures of neurons and glial cells, respectively. The tissues were dissociated with 0.25% Trypsin (Gibco) and 0.1 mg/ml DNaseI (Roche) for 15 min at 37 °C. The cells were plated in poly-D-lysine-coated T25 flasks or 24-well plates at ~80,000 cells per cm², in the seeding medium consisting of DMEM-F12 (Corning), 10% FBS (Gibco), and 1% Antibiotic-Antimycotic Solution (Corning). For neuronal cultures, the media was exchanged to Neurobasal (Gibco), 2% B-27 (Gibco), 1% Antibiotic-Antimycotic Solution (Corning) and 0.5 mM Glutamax (Gibco) 1 day after plating. Mature neurons at 21 days in culture have been utilized for the RNAseq. For glial cultures, the flasks were shaken (200 rpm at 37 °C) three times overnight to remove microglia, and astrocytes trypsinized and further cultured in 24-well plates. For microglia cultures, the media was supplemented with the recombinant M-CSF mouse protein (10 ng/ml; Gibco). Floating microglial cells were collected from the conditioned media by gentle spin (300×g, 10 min), and re-plated in 24-well poly-D-lysine-coated plates at ~100,000 cells per cm². For human primary neuroglial cultures, fetal cortical tissues (gestational age 18 weeks) were provided by Advanced Bioscience Resources, Inc. (Alameda, CA), dissociated with papain (12 U/ml; Worthington), seeded with neuronal media plus 2% FBS (Gibco), and cultured as mouse neurons[63]. Neuronal cultures at 30 days in vitro have been utilized for the RNAseq. Human (HBEC) and mouse (MBEC) primary brain microvascular endothelial cells were purchased from Cell Biologics, IL (Catalog# H-6023; Lot# 021514F14 and Catalog# C57-6023; Lot# 070613T2MP, respectively) and cultured accordingly to the manufacturer. All animal experiments have been approved by the Harvard Medical Area Standing Committee on Animals.

**Fractionation and RNA isolation.** Approximately 100 ml conditioned medium was used as input for exRNA isolation. Conditioned media was centrifuged at 300×g, 4 °C for 10 min, following the additional centrifugation at 2000×g, 4 °C for 15 min, to remove cells and cell debris. To monitor EVs, the samples were diluted in DPBS and examined using the Nanoparticle Tracking Analysis system (NanoSight LM10; Malvern Instruments, UK), and EV concentrations were quantified within the optimal linear range ($2–10 \times 10^8$ particles per 1 ml).

For RNA preparation, 5 µl of SUPERase In RNase Inhibitor (Ambion) was added to the supernatants per 10 ml media. The media was then filtered

sequentially through the 2 μm filter (GE Healthcare, UK), 0.8 μm filter (EMD Millipore, MA), and 0.22 μm filter (EMD Millipore), with no/minimal pressure applied. The filtrate was split to 15 ml per sample and further filtered through the 0.02 μm filter (GE Healthcare) with up to 75 psi pressure applied. To facilitate the 0.02 μm filtration, a mechanical syringe pump was designed and manufactured (Supplementary Fig. 2). Upon filtration, each filter was washed with 1 ml DPBS (Corning), and the corresponding fractions were lysed with 600 μl lysis solution of the miRCURY RNA Isolation Kit—Cell & Plant (Exiqon, Denmark). The fractions collected on 0.02 μm filters were lysed with 900 μl lysis solution. The last flow-through fractions of 0.02 μm filters were pooled together (up to 30 ml) and concentrated ~60 times using 3 kDa Amicon Ultra Centrifugal Filters (EMD Millipore) at 4000×g, 4 °C, for 60 min. The concentrates were collected and lysed with six volumes of the same lysis solution (Exiqon). Total RNA was then isolated from all fractions as recommended by miRCURY protocol, with on-column DNase treatment (Qiagen, Germany). The corresponding 1.2 ml of the source neurospheres were span down at 300×g, 4 °C for 5 min, and total cellular RNA was isolated from them and analyzed in parallel. The same protocol was carried out for RNA isolation from fresh media, with 500 ml media input. For RNA isolation from primary cells cultured in 24-well plates, the cells were lysed with 350 μl lysis solution per well. The concentrations of cellular and extracellular RNA were determined using the NanoDrop 2000 Spectrophotometer and Quant-iT RiboGreen RNA Assay Kit (Thermo Fisher Scientific), respectively. The RNA quality was examined using Agilent 2100 Bioanalyzer (Agilent, CA) and the RNA Integrity Number (RIN) estimated.

**Transmission electron microscopy**. The material collected on the filters was resuspended using DPBS (Corning), and further pelleted by 100,000×g UC for 80 min at 4 °C. The material diluted in DPBS was added to a glow-discharged carbon-coated grid. The grids were washed with distilled water, stained with 0.75% uranyl formate, examined using Tecnai G² Spirit BioTWIN microscopy (FEI, OR), and images recorded by the AMT 2k CCD camera (Advanced Microscopy Techniques, MA) at the Harvard Medical School EM Facility.

**RNA sequencing**. Two sets of spike-in RNAs were added to the samples prior to library preparation: ERCC RNA Spike-In Control Mixes (Ambion; 0.02 μl per 1 μg total RNA), and miRCURY Spike-in kit, part 1, with UniSp2's final concentrations of 1.25 and 5.0 fmol per 1 μg of total RNA for cellular and extracellular RNA, respectively. Total RNA, either 40–200 ng of exRNA, or 2 μg of cellular RNA, was rRNA-depleted using the Ribo-Zero rRNA Removal Kits (Illumina, CA). One quarter of the rRNA-depleted RNA was fragmented to 100–500 nt using the 5× First-Strand Buffer (Clontech, CA), and utilized for the long RNA library construction by SMARTer Stranded RNA-Seq Kit (Clontech). The remaining 75% of the rRNA-depleted RNA was treated sequentially with Tobacco Acid Pyrophosphatase (TAP; Illumina) and T4 Polynucleotide Kinase (T4PNK; New England Biolabs, MA) to create more uniform 5′- and 3′-ends for various classes of transcripts. The RNA was then used as input for the NEBNext Multiplex Small RNA Library Prep Set for Illumina (New England Biolabs), with size selection of 15–65 nt inserts for small RNA libraries. The quality of libraries was examined using the Agilent DNA 1000 kit at the Agilent 2100 Bioanalyzer instrument, and cDNA quantified by qRT-PCR. The libraries were sequenced on HiSeq 2000 (Illumina) with single read 50 cycles by Beijing Genomics Institute (BGI, China).

**Reads annotation**. Sequencing reads were treated using the BGI pipeline that included multiple filtering steps, as follows: (1) removing reads with adapters; (2) removing reads with >10% of unknown bases; and (3) removing low-quality reads (sequencing quality <10). After filtering, the remaining clean reads were subjected to the bioinformatics analysis. Generally, 20 and 10 M clean reads were generated per long RNA library and small RNA library, respectively. The clean reads produced from the long RNA libraries were first mapped to the Homo sapiens rRNA database using the SOAPaligner/SOAP2 short read alignment software, to remove the remaining rRNA reads. The non-rRNA reads were used to perform the transcriptome assembling and quantification. First, non-rRNA reads were mapped to human reference genome hg19 using an improved version of TopHat2, which aligns reads across splice junction without relying on gene annotation. Default -g/--max-multihits was used to allow up to 20 multimapping. Next, the reads mapped to genome were assembled using Cufflinks. Reference Annotation Based Transcript (RABT) assembly was performed with the reference gene annotation to compensate incompletely assembled transcripts caused by read coverage gaps in the regions of reference gene. The set of transfrags generated was then compared with the reference transcripts to remove transfrags that were approximately equivalent to the whole or a portion of a reference transcript. After the assembling, the whole parsimonious set of transcripts was obtained. These transcripts were blasted with the NONCODE database using the filter set (identity >0.9 and coverage >0.8) to identify known long non-coding RNA. Then, the rest of assembled transcripts were aligned to the reference annotation utilizing Cuffcompare. Thereafter, Cuffmerge was utilized to merge several assemblies from different samples together, which automatically filtered out a number of transfrags that probably were artifacts and produced a single annotation file for downstream gene expression analysis. Two mismatches were allowed for annotation generally, and only one mismatch was allowed for the ERCC spike-ins. mRNA and lncRNA expression analyses were

performed by Cuffdiff with parameter -u/--multi-read-correct and normalized based on the ERCC spike-in controls using the cyclic loess robust local regression.

Clean reads produced from the small RNA libraries were first aligned to carrier RNA (Enterobacteria phage MS2) of miRCURY Spike-in kit with bowtie (parameter: -v 2 -l 7 --all). The unmapped reads were aligned against human rRNA sequences with bowtie (parameter: -v 2 -l 7 --all) to remove the remaining rRNA reads. The remaining reads were mapped to human miRNA precursors (miRBase V19) with bowtie (parameter: -v 1 --all), and the miRNA precursors reads were separated according to the mapping position. The mapped reads with more than 10% mismatched nucleotides were excluded. The remaining reads were mapped to spike-ins (UniSp2, UniSp4 and UniSp5; parameter: -v 1 --all), and other non-coding RNA classes including tRNAs (gtRNAdb), piRNAs (RNAdb), snoRNA (snoRNA-LBME-db), scRNAs (Genbank), and others (Rfam database) (parameter: -v 2 --all). Finally, the remaining reads were mapped to the hg19 human genome, and the reads mapped to exons, introns, intergenic regions, and repeats (parameter: -v 0 --all) identified. The SAMtools was used to calculate the reads depth for each base position and R package barplot was used to draw the corresponding bar plot.

**Data analysis**. Based on the known amounts of spike-in RNAs, the read count for each RNA species was first normalized to spike-ins' reads, to quantify the absolute amounts in fmol per μg of total RNA. Next, the obtained values were corrected to the corresponding fresh media as the blank control, using the equation shown in Supplementary Fig. 6. For the analysis of class composition, the total abundance of corrected non-rRNA was used for normalization between the samples. To compare heterogeneity of the samples, the sum of squared errors ($\chi^2$ value) of species composition was calculated using MS Excel Macro (available in Supplementary Data 3). To estimate the inequality of abundance among all RNA species in one class, the evenness factor, Gini coefficient, and traditional pre-set evaluations were calculated using a MS Excel Macro (available in Supplementary Data 4). The hierarchical clustering analysis was performed using the MultiExperiment Viewer (Dana-Farber Cancer Institute, MA). The principle component analysis (PCA) was performed and visualized with R package rgl. miRNA targets were determined based on starBase v2.0, with at least three supporting CLIP-Seq experiments[33]. Pathway analysis of target mRNAs was performed using the Ingenuity Pathway Analysis (IPA; Qiagen). Venn diagrams were plotted using Venny 2.1.

**Long RNA reverse transcription PCR**. Maxima Reverse Transcriptase (100U; Thermo Fisher Scientific) was used to reverse transcribe 20 ng RNA with both oligo (dT) and random hexamers in a 10 μl reaction system. Next, 0.5 μl cDNA was used in 10 μl PCR reactions based on Phire Hot Start II PCR Master Mix (Thermo Fisher Scientific) and primers (0.5 μM each), and amplified using a touchdown program with either 15 or 70 s extension step for short or long amplification, respectively. The PCR products were examined on 0.8–1.2% agarose gels (Thermo Fisher Scientific). Primer sequences are provided in Supplementary Table 3.

**Quantitative reverse transcription PCR**. Generally, for small RNA qRT-PCR, 10 ng of total RNA was used in 10 μl reverse transcription reaction with Universal cDNA Synthesis kit II (Exiqon). The cDNA was diluted 80 times, and 4 μl was used in 10 μl qPCR reactions using the ExiLENT SYBR Green master mix and custom-designed LNA primers (Exiqon). For mRNA qRT-PCR, 10 ng of total RNA was used in 10 μl reverse transcription reaction with PrimeScript RT Master Mix (Takara, Japan). One microliter cDNA was used in 10 μl qPCR reactions using the ExiLENT SYBR Green master mix and synthesized primers (Supplementary Table 4; Eton Bioscience, CA). The qPCR reactions were run on a ViiA 7 instrument (Thermo Fisher Scientific) in duplicates. The specificity of qPCR products was verified by the presence of a single peak at the melting curves.

**Immunoblotting**. For protein analysis, cellular and extracellular fractions were lysed using the modified RIPA buffer containing 2% SDS, 1% sodium deoxycholate and 3 M urea. The RNP fractions were BSA-depleted using the Aurum Affi-Gel Blue Mini Columns (Bio-Rad, CA), and concentrated using Pierce SDS-PAGE Sample Prep Kit (Pierce, MA). Protein concentrations were quantified using the Micro BCA Protein Assay Kit (Pierce). Equal amounts of the total protein (50 μg) were loaded per lane in Novex WedgeWell 14% Tris-Glycine Mini Gel (Thermo Fisher Scientific). Proteins were transferred to 0.45 μm PVDF membrane (Thermo Fisher Scientific). After blocking with 5% (wt/vol) fat-free milk in Tris-buffered saline with 0.075% Tween-20 (TBST), membranes were incubated with 1:1000 diluted primary antibodies (Flotillin-1 #18634S, CD9 #13174S, Integrin β1 #9699S, HSP90 #4874S, La #5034S, NPM #3542S, Ago2 #2897S, Alix #2171S, and PCNA #13110S from Cell Signaling Technology, MA; Ro60 #AV40534 from Sigma-Aldrich; ANG #sc-74528 from Santa Cruz Biotechnology, TX) overnight at 4 °C. The membranes were washed and incubated with horseradish peroxidase–conjugated secondary antibodies (#7074S and #7076S from Cell Signaling Technology, 1:2000 dilution) for 1 h at room temperature. The blots were developed by the Amersham ECL Reagent (GE Healthcare) and, if required, stripped using One Minute Western Blot Stripping Buffer (GM Biosciences, CT). Uncropped scans of blots are shown in Supplementary Fig. 19.

**Transwell cocultures of GSC and astrocytes**. Mouse cortical astrocytes established from WT or miR-21 KO P1 pups, passage 1 or 2, were cultured in 24-well plates. Small GBM8 neurospheres grown in GSC conditions were transferred to the upper chamber of the Millicell hanging insert with 1.0 µm pore size (Millipore), and co-cultured with the pre-established astrocytes. The co-culture media was adjusted to contain DMEM-F12 (Corning), 1× B-27 supplement (Gibco), 0.5× N-2 (Gibco), and 1% Antibiotic–Antimycotic Solution (Corning), enabling co-culturing of healthy astrocytes and GSC neurospheres over a period of at least 6 days. The corresponding control "mono-cultures" of astrocytes were cultured in the identical media under the same conditions. Three days later, astrocytes were washed with cold DPBS (Corning) three times, and cellular RNA isolated as in other experiments.

**Fractionation of cerebrospinal fluid**. Cerebrospinal fluid (CSF)samples from four patients diagnosed with primary GBM were collected during surgery, with informed consent obtained according to the appropriate protocol, approved by the UC San Diego Institutional Review Board. The samples were centrifuged at 1500×g at room temperature for 10 min, immediately after collection to remove cells. The supernatants were filtered through the 0.8 µm filters (Millipore), aliquoted into cryotubes (Thermo Fisher Scientific), and stored at −80 ℃ for 12–24 months. The aliquots of 3 ml were thawed on ice, and used to isolate MV, exosomes, and RNP fractions by sequential filtration, as described above.

**Statistics**. Values are given as mean ± SEM. Numbers of experimental replicates are given in the figure legends. When two groups were compared, significance was determined using an unpaired two-sided $t$-test. Before performing $t$-test, normal distribution was verified by one-sample Kolmogorov–Smirnov test and equality of variances was verified by Levene's test using SPSS (IBM, NY). A $p$-value < 0.05 is considered as statistical significance. All bars represent mean ± SEM.

**Data availability**. RNAseq fq files and processed data are available through GEO under GSE93143. MS Excel Macros used to calculate inequality and heterogeneity are available in Supplementary Data 3 and 4. All relevant data are available within the Article and Supplementary Files, or available from the authors upon request.

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

## Acknowledgements

This work was supported by the U19 CA179563 grant and U19 administrative supplement through the NIH Common Fund and the Office of Strategic Coordination/Office of the NIH Director (A.M.K. and X.O.B.), R21 NS098051 (A.M.K.), and CA069246 (X.O.B.) grants. We thank Robert Kitchen, Sai Lakshmi Subramanian, Alain Charest, Steven Gould, Thorsten Mempel, and the RNAseq working group of the Extracellular RNA Communication Consortium for valuable insights on the data analysis. We thank Maria Ericsson for TEM, Hongjun Wang for designing and manufacturing syringe pumps, Pavel Ivanov for sharing ANG antibody, Oleg Butovsky for sharing miR-21 knockout mice, and Erik J. Uhlmann for reviewing the manuscript. We thank members of Krichevsky and Breakefield laboratories for helpful discussions.

## Author contributions

A.M.K. conceived the study; Z.W. and A.M.K. designed experiments; Z.W. performed experiments; S.S., R.E.F., Y.W., R.R. and L.B. assisted with experiments; A.B. and J.W. performed bioinformatics analysis; X.O.B. contributed to data analysis; F.H., C.C. and B.C. provided CSF samples; Z.W. and A.M.K. analyzed the data and wrote the paper. All authors revised and approved the manuscript.

## Additional information

**Competing interests:** X.O.B. is a consulting member of the Scientific Advisory Board of Evox Therapeutics, Ltd. and Exocyte Therapeutics, Ltd. The remaining authors declare no competing financial interests.

