## [Peer Review File · Nature Communications]

Reviewers' comments:

Reviewer #1 (Remarks to the Author):

This manuscript presents a study of extracellular RNA released by cultured primary GBM cells. The authors analyzed RNA expression profiles of samples collected from extracellular vesicles and ribonucleoprotein. They also compared RNA expression profiles of intracellular samples to those derived from extracellular fractions. The study provides a comprehensive view of exRNA expression profiles of GBM cells. Importantly, usage of spike-in controls allowed quantification of the amount of exRNAs in each fraction. Data described in this manuscript are very interesting. However, I have a few concerns:

It is not clear whether some of the fractions (e.g., MVs) are contaminated by cellular RNA that may have been released by ruptured cells during the isolation procedure. Since this study highlighted comparison of intra- vs. extracellular RNA profiles, it is critical to evaluate and exclude the possibility of cellular RNA contamination.

The authors mentioned that the EV isolation procedures may induce fragmentation of RNA. It is important to know how much fragmentation there is in general, on the global scale. Since they used spike-in controls, can the authors evaluate the fragmentation level using these spike-ins?

The authors described the observed heterogeneity of RNA composition among cultures. However, it is not clear how much of the heterogeneity was derived from technical variation. It is important to carry out biological replicates of the same patient culture and repeat all experimental procedures to demonstrate variability due to technical variation in EV isolation, RNA isolation, RNA-seq etc. In general, biological replicates are needed to evaluate how reproducible these protocols are.

In Figure 4a, can the authors provide absolute amount of each type of RNA in each fraction? Such data will be very helpful in understanding the quantitative distribution of exRNA in fractions.

In Figure 4d, was this analysis corrected by length, given the fact that 3' UTR, 5'UTR and CDS have different length distributions?

It is not clear whether small RNA and long RNA reads were mapped by requiring uniqueness in mapping. The authors should expand the Method section to include more details.

The authors attempted to compare intra- and extracellular RNA expression profiles. Would it be possible to use more traditional methods to compare differential expression (such as edgeR or DEseq)? A method that can identify specific genes that are differentially expressed within cells vs. outside of cells (or across fractions) will be more powerful. In addition, such analyses will allow further examination of functional categories of RNA that are enriched in the extracellular fractions in particular.

Recent literature reported discovery of circular RNAs in extracellular RNA space. Can the authors analyze their data to determine if circular RNAs are present?

Reviewer #2 (Remarks to the Author):

The manuscript by Dr. Krichevsky and colleagues describe a detailed comparative analysis of

extracellular RNA (exRNA) in microvesicles (MVs), exosomes, and ribonucleoprotein complexes (RNPs) secreted from glioblastoma cells. The in-depth analysis of RNA composition in extracellular vesicles has been overlooked in the past, and the results of this study bring useful insight for the field. Using RNA-seq data, the authors clearly show the different characteristics and categories of RNA species in extracellular fractions and source cells.

While the authors provide extensive data analysis on RNA-seq data obtained from MV, exosomes, and RNPs, the study lacks in conclusive evidence that show the physiological significance of their findings. Overall, the significance of this study can be improved greatly by additional functional analysis of RNA transfer, uptake, and activity in exRNA recipient cells. In this present form, it is difficult to understand the implications of this study.

Major comments:

1. The authors state that sequential filtration is superior to other conventional methods such as ultracentrifugation, precipitation and fractionation, and state that this protocol achieves “better separation of EVs and RNPs, higher yield of RNA”. This statement lacks evidence and the authors should indicate references. It may be important to explain how different methods would result in different RNA yields, as it could be important in the assessment of subsequent studies. In particular, the authors should show evidence for Fig. 1b.
2. The electron microscopy image of MV in Fig. 2a should be replaced. The size of particle in Fig. 2a may not be representative of a MV (0.2-0.8 μm).
3. The enrichment of miR-122 in exRNA before and after media correction is unclear in page 6. The authors should clarify whether exRNA fractions in Fig. 3b and 3c originates from the B-27 supplement.
4. What do the authors mean in page 8, “Such changes in inequality, both a decrease (e.g. case III in Fig. 5b) and an increase (e.g. case IV in Fig. 5b), suggest the selectivity of exRNA released by cells”? The authors need to elaborate on the importance of the model in Fig. 5b, as it does not provide additional support to the text.
5. The authors claim that exosomal miRNA has high functional potential upon uptake by recipient cells. However, the term “uptake” is not appropriate, as it denotes actual functional transfer to recipient cells. The study lacks functional data regarding RNA transfer and activity in recipient cells. The uptake of miRNA by recipient cells requires functional assays.
6. There is little evidence to show that the mRNA targets are validated in Fig. 8. In addition to this, the statement in page 19, “We identified miRNAs with the highest potential for affecting normal brain cells...” would be an overstatement without supportive evidence.
7. The authors claim that the results of this study will provide a “catalog of diverse vesicular and extravesicular exRNA species useful for biomarker discovery”. In addition to this, they state repeatedly that MVs provide a more accurate read-out of the source cell content than exosomes and could be utilized for identification of biomarkers. This claim could be strengthened by additional evidence that show RNA profiles (or at least a particular read-out) of MVs isolated from patient samples such as blood, cerebrospinal fluid, urine.
8. A major concern is the stability of RNA from MVs, exosomes, and RNP complexes. MVs may present a highly promising source of liquid biopsy biomarkers, but can they still be utilized from stored patient samples? What should be taken into consideration when using patient samples for detecting RNA from MVs over relatively stable miRNA in exosomes?
9. Sequential filtration, in addition to the procedures that were used for RNA extraction and library preparation may have caused RNA degradation and fragmentation. How would the authors consider the loss of intact RNA by such methods?

Minor comments:

1. Fig. 2c is not described in the text.
2. Overlooked typo in page 5 “The caveat s that this procedure...”
3. Inappropriate use of italics used in page 6 “Whereas in 3 out of 4 GSC cultures cellular mRNA reads accounted for...”
4. Fig. 4c does not seem to support or agree with Fig. 4b. The length distribution of RNA in MV,

exosome and RNP fractions do not seem to differ in Fig. 4c, which is conflicting with Fig. 4b, which shows the different lengths of RNA in the extracellular fractions. The authors should modify the figure to clarify this point.

5. What do the scale bars represent in the heat map cluster analysis in Fig. 6a, 6d and 6f?

6. Fig. 6b requires correction. The angle of the PCA plot is not appropriate.

7. Fig. 7c is unclear. What do the authors mean in their claim that the mapping of Y1 is "precisely processed"? The figure is unclear, and needs more contextual information.

8. Some terms are vague or not appropriate in the scientific context, and should be replaced with appropriate terms that hold scientific meaning and could be understood in the context of this paper. E.g. page 13 "cell ecosystems", "indigenous cancer cells", "transcriptional programs".

9. The use of terms "RNA species", "RNA classes", "RNA fractions", should be consistent throughout the text.

10. Could apoptotic vesicles also be contained in the MV fraction?

Reviewer #1 (Remarks to the Author):

This manuscript presents a study of extracellular RNA released by cultured primary GBM cells. The authors analyzed RNA expression profiles of samples collected from extracellular vesicles and ribonucleoprotein. They also compared RNA expression profiles of intracellular samples to those derived from extracellular fractions. The study provides a comprehensive view of exRNA expression profiles of GBM cells. Importantly, usage of spike-in controls allowed quantification of the amount of exRNAs in each fraction. Data described in this manuscript are very interesting. However, I have a few concerns:

It is not clear whether some of the fractions (e.g., MVs) are contaminated by cellular RNA that may have been released by ruptured cells during the isolation procedure. Since this study highlighted comparison of intra- vs. extracellular RNA profiles, it is critical to evaluate and exclude the possibility of cellular RNA contamination.

Response:

We agree that purity of the fractions is critical for comparison between intra- and extracellular RNA, especially considering the high sensitivity of technologies utilized in our study (i.e. RNAseq). We used the conventional protocol of two-step centrifugation (300g 10min and 2,000g 15min) to remove the cells, followed by additional filtrations through 2.0um and 0.8um pores, preceding the MV isolation (Fig.1a). This multistep procedure is more stringent than most of others previously reported, and it ensures no cell/cell debris contamination of the extracellular fractions. Remarkable differences in the levels of several protein markers between cellular and extracellular fractions validate the lack of cellular contamination (Fig. 2b).

We have added the following sentence on Page 5 of the revised manuscript: “Distinct profiles of several protein markers exhibited by cellular and extracellular fractions served to confirm the purity of fractions and the lack of cellular contamination in the MV and other extracellular fractions (Fig. 2b).”

The authors mentioned that the EV isolation procedures may induce fragmentation of RNA. It is important to know how much fragmentation there is in general, on the global scale. Since they used spike-in controls, can the authors evaluate the fragmentation level using these spike-ins?

Response:

We appreciate this important point raised by the Reviewer and apologize for not explaining our data more clearly. The data indicate that the ratio between fragments and parent full-length transcripts is generally higher in smaller vesicles/particles than in the cells. The procedure of sequential filtration is unlikely to cause RNA fragmentation, since the shear force applied is low and insufficient for rupturing the vesicles and affecting the RNA. Nevertheless, we attempted to confirm this point experimentally. We could not apply the analysis of spike-ins suggested by the Reviewer, since they were added after the RNA isolation (mostly to control for RNAseq bias). However, we have now added a new experiment demonstrating that total exRNA isolated from the unfractionated conditioned media (cell-free 2,000g-supernatant), not subjected to the filtration procedure, exhibited the fragmentation rates similar to those in filtrated exosomes and RNPs. As

shown in Fig. S13, the fragment to full-length ratios were much higher in this unfractionated exRNA than in the cellular RNA, indicating that the fragmentation is a feature of exRNA rather than the artifact of the filtration procedure. In addition, we should note that most common extracellular RNA fragments were highly specific (e.g. 30nt 5' fragments of Y RNA and tRNA, Fig. 6c and Fig. S12). This data reinforces the idea that it is RNA processing and selection, rather than unspecific degradation that accounts for the profile of exRNA.

The authors described the observed heterogeneity of RNA composition among cultures. However, it is not clear how much of the heterogeneity was derived from technical variation. It is important to carry out biological replicates of the same patient culture and repeat all experimental procedures to demonstrate variability due to technical variation in EV isolation, RNA isolation, RNA-seq etc. In general, biological replicates are needed to evaluate how reproducible these protocols are.

Response:

We agree with the Reviewer that assessment of technical variability and reproducibility of the methodology is critically important, although including biological replicates in high-content studies such as those based on RNAseq is still prohibitively expensive, and thus rarely performed. To address the concern of technical reproducibility, and also of biological variability between low-passage cultures derived from the same patient, we have added a new analysis that compared two independent RNA preps from different passages of GBM8 neurospheres. Arguing that rRNA depletion should not affect the overall results of the RNAseq, we performed the analysis of one sample including the rRNA depletion step (as for all other experiments in this study), whereas the cDNA library preparation on the second culture has omitted this step. As shown in Fig. S10, despite the non-identical library construction procedures performed for two GBM8 cultures, their exRNA profiles are much more concordant (Fig. S10a) than the pairwise-compared profiles of exRNA released by GSC cultures established from different patients, even though those were run in parallel, under the same technical protocol (Fig. S10b-d).

This experiment further supports our conclusion that the increased heterogeneity of extracellular fractions has a biological rather than technical origin/cause. The corresponding text is now added on page 9 of the revised manuscript.

In Figure 4a, can the authors provide absolute amount of each type of RNA in each fraction? Such data will be very helpful in understanding the quantitative distribution of exRNA in fractions.

Response:

We agree with the reviewer that the lack of such quantitative assessment represents a critical gap in the field of exRNA. The absolute quantification is, indeed, challenging due to the technical limitations of current methodologies used for EV analysis. For example, NTA/NanoSight, the most common among them, does not provide accurate and reproducible quantification of EVs, nor does it discriminate between MVs and exosomes. There are also no current methods established for the absolute quantification of extra-vesicular RNPs. Considering these factors, as well as the observation that RNA secretion varies significantly among GSC cultures and thus the absolute

values cannot be presented as a bar plot, we summarized this information in the revised Table 1. The Table includes quantitative information for multiple RNA classes and characteristic individual RNA species found in EVs (MVs and exosomes collectively).

In Figure 4d, was this analysis corrected by length, given the fact that 3' UTR, 5'UTR and CDS have different length distributions?

Response:

We apologize for the brief description of the analysis, and now provide a more detailed explanation. The data for each mRNA region (CDS, 5'-UTRs, and 3'-UTRs) in exRNA fractions were normalized to the corresponding data for cellular RNA, and presented as, for example, the logarithm of the alignment rate ratios [3'-UTR in MV / to 3'-UTR in cell]. Therefore, the variability in length distribution among 5'UTR, 3'UTR, and CDS does not cause bias. This explanation has been added to Page 8 of the revised manuscript.

It is not clear whether small RNA and long RNA reads were mapped by requiring uniqueness in mapping. The authors should expand the Method section to include more details.

Response:

Since our goal was to characterize the exRNA fractions without focusing on miRNA or any other RNA class, we have had to utilize many reference databases with partly overlapping contents. Therefore, we have not required the uniqueness of the mapping. Multi-mapping up to 20 was allowed for the long RNA annotation and was not limited for the small RNA annotation. We have now included more details on this analysis in the Method section.

The authors attempted to compare intra- and extracellular RNA expression profiles. Would it be possible to use more traditional methods to compare differential expression (such as edgeR or DEseq)? A method that can identify specific genes that are differential expressed within cells vs. outside of cells (or across fractions) will be more powerful. In addition, such analyses will allow further examination of functional categories of RNA that are enriched in the extracellular fractions in particular.

Response:

Traditional methods used for the differential expression analysis, such as edgeR and DEseq, are based on the assumption of negative binomial distribution, which addresses the nature of sequencing reads better than Gaussian distribution. However, they both require read counts (integer numbers) as input. As we demonstrated before [1], and discussed in this manuscript, media-derived RNA contaminates the cell-derived exRNA, and such contamination must be corrected for the analysis of enriched exRNA species. Therefore, our analytical pipeline includes the correction step that transforms the RNAseq data into the abundance data ("fmol per ug total RNA"), and is therefore not suitable as input for the edgeR or DEseq methods.

To identify transcripts that are differentially expressed in exRNA fractions vs. parental cells (or in other words, those enriched in exRNA), we calculated a log₁₀-transformed fold change for each RNA species in each extracellular fraction vs. GSC culture. We then applied a t-test to examine the fold changes in four GSC cultures. The newly included Table S5 lists, for each extracellular fraction,

the exRNA species that fulfill the following enrichment criteria: mean of log₁₀-transformed fold change more than 1.0 (equivalent of 10-fold enrichment) and p value < 0.05. Considering the loose p value cut-off applied, the difference of p value caused by different statistical models (our method versus edgeR and DEseq) should have no significant effects.

Recent literature reported discovery of circular RNAs in extracellular RNA space. Can the authors analyze their data to determine if circular RNAs are present?

Response:

The design of our library construction and sequencing (single end 50 bp reads) was, unfortunately, not optimized for the annotation of the recently recognized circular RNA species that usually range between 200-800 nt [2] and are sequenced with pair end 100 bp reads [3, 4]. Nevertheless, we used exceRpt small RNA-seq pipeline of the Genboree Workbench [5] to attempt the annotation of circular RNAs within our dataset. Overall, less than 0.006% of the reads were mapped to circular RNAs in our small and long RNA libraries, possibly due to the suboptimal procedure, and/or their low levels in GSC cellular and extracellular RNAs. The detected circular RNAs are listed in the Table S1, and the corresponding text is added to page 7 of the Results. We don't find this data very useful, and could either keep or remove it from the Manuscript, per Editorial decision.

Reviewer #2 (Remarks to the Author):

The manuscript by Dr. Krichevsky and colleagues describe a detailed comparative analysis of extracellular RNA (exRNA) in microvesicles (MVs), exosomes, and ribonucleoprotein complexes (RNPs) secreted from glioblastoma cells. The in-depth analysis of RNA composition in extracellular vesicles has been overlooked in the past, and the results of this study bring useful insight for the field. Using RNA-seq data, the authors clearly show the different characteristics and categories of RNA species in extracellular fractions and source cells.

While the authors provide extensive data analysis on RNA-seq data obtained from MV, exosomes, and RNPs, the study lacks in conclusive evidence that show the physiological significance of their findings. Overall, the significance of this study can be improved greatly by additional functional analysis of RNA transfer, uptake, and activity in exRNA recipient cells. In this present form, it is difficult to understand the implications of this study.

Response:

We agree that, indeed, the major significance of our study is in providing the in-depth characterization and quantification of the exRNA landscape. As members of the NIH-funded Extracellular RNA Consortium, we are well-aware that the field is in critical need of such datasets. Other similar descriptive studies (e.g. miRNA expression atlas by Tuschl group, Cell, 2007 [6]; and promoter-level expression atlas, Nature, 2014 [7]), published in high-impact journals and cited by hundreds/thousands reports, truly set the ground for their respective fields. Although the newly added data (Fig. 8d,e) and previous reports on miRNA (van der Vos KE et al, Neuro-Oncol [8]) and mRNA (Lai CP et al, Nat Commun [9]) demonstrate the horizontal miRNA and mRNA transfer between cells and support their functional impact on the recipient cells, this manuscript highlights the complexity of exRNA "space" comprising thousands individual RNA species, protein-coding,

and non-coding. We believe that accurate, unbiased, and controlled dissection of the functionality of those individual species will be a long-term task that is beyond the scope of this manuscript. We further expand this discussion of this topic on page 20 of the revised manuscript.

In addition to the functionality of this RNA transfer, the composition of exRNA has become an important source of biomarkers for many disease states [10], including GBM [11].

Major comments:

1. The authors state that sequential filtration is superior to other conventional methods such as ultracentrifugation, precipitation and fractionation, and state that this protocol achieves “better separation of EVs and RNPs, higher yield of RNA”. This statement lacks evidence and the authors should indicate references. It may be important to explain how different methods would result in different RNA yields, as it could be important in the assessment of subsequent studies. In particular, the authors should show evidence for Fig. 1b.

Response:

In addition to the data shown previous Fig. S1, we have now included the direct comparison of ultracentrifugation with sequential filtration (new Fig. 1b), and added several references supporting our statement. The new experiment analyzed 8 aliquots of conditioned media in parallel, four of which were processed by ultracentrifugation and other four by filtration. Based on the NanoSight analysis (shown in Fig. 1b and Fig. S1) and previous reports [12, 13], ultracentrifugation does not pellet the exosomes completely, leading to the 1) reduced RNA yield in the exosomal preparations, and 2) contamination of the RNP fraction by the remaining vesicles. In contrast, vesicles were barely detectable in the flow-through of the 20nm filters by NanoSight analysis, (Fig. 1b and Fig. S1), and undetectable by TEM (Fig. 2a). This data further supports the observation that sequential filtration provides better separation between the EVs and RNPs, and higher RNA yield than ultracentrifugation.

Other characteristics listed in Fig.1c are primarily based on simple physics. For example, the pressure on vesicles during ultracentrifugation = the media density (about 1 g/cm³) * gravity force (100,000 * 9.8 N/kg) * depth (about 9 cm) = 12,792 psi; while the pressure during filtration is less than 100 psi. Therefore, we stated that pressure on EVs was reduced at least 100 times using filtration.

2. The electron microscopy image of MV in Fig. 2a should be replaced. The size of particle in Fig. 2a may not be representative of a MV (0.2-0.8 μm).

Response:

We have replaced the MV image in Fig. 2a with a more representative image.

3. The enrichment of miR-122 in exRNA before and after media correction is unclear in page 6. The authors should clarify whether exRNA fractions in Fig. 3b and 3c originates from the B-27 supplement.

Response:

To address this question, we have isolated total RNA from components of the fresh media (including B27, N2, and supplementary growth factors), and analyzed them for miR-122 levels using qRT-PCR. As expected, miR-122 was fairly abundant in B-27 but undetectable in other components (Fig. S5). We have modified our text as following: "... miR-122 was highly enriched in the GSC exRNA, relative to its intracellular level, with up to 1,500-fold enrichment in the exosomes before media correction (Fig. 3c), consistent with previous reports [18, 19]. However, miR-122 reads were also highly abundant in the FM-derived RNA, due to the abundance of miR-122 in the B-27 supplement (Fig. S5)."

4. What do the authors mean in page 8, "Such changes in inequality, both a decrease (e.g. case III in Fig. 5b) and an increase (e.g. case IV in Fig. 5b), suggest the selectivity of exRNA released by cells"? The authors need to elaborate on the importance of the model in Fig. 5b, as it does not provide additional support to the text.

Response:

The topic of random versus selective RNA incorporation into different exRNA complexes has been debated in the exRNA field since RNA was detected in EVs. Previous evidence for selective incorporation were mainly based on the observations that a subset of miRNAs was enriched in exRNA in comparison to cellular RNA. No global evaluation on all RNA classes/species has been performed to address this dilemma. Our inequality analysis provides new evidence to support the selectivity of incorporation of exRNA into MVs, exosomes, and RNPs (Fig. 5a). Since this type of analysis is rarely applied in this context, we felt that Fig. 5b served to illustrate our conclusions. However, we agree with the Reviewer that this panel is unnecessary and have removed it from the revised manuscript. We have also revised the text at page 9 accordingly, to clarify our conclusions.

5. The authors claim that exosomal miRNA has high functional potential upon uptake by recipient cells. However, the term "uptake" is not appropriate, as it denotes actual functional transfer to recipient cells. The study lacks functional data regarding RNA transfer and activity in recipient cells. The uptake of miRNA by recipient cells requires functional assays.

Response:

Whether RNA transfer between the cells can exert functional effects is a key question that has yet to be addressed convincingly by the entire community of exRNA investigators, although a number of studies supports such functional transfer (see response above). The goal of our current manuscript was not to answer this question, but to draw attention to the complexity of exRNA. Nevertheless, in the revised version of the manuscript we provide the data supporting such functional transfer, while emphasizing the challenges associated with its detection.

Specifically, to examine the transfer and functionality of GSC-derived miRNAs, we established 3-D Transwell co-cultures of GBM8 neurospheres with primary mouse astrocytes, a system which is more physiological than the commonly used exposure of recipient cultures to the super-concentrated EVs derived from donor cells. In line with the low copy number of individual miRNAs per EV, we were unable to detect significant elevation of either miR-21 or miR-10b (key tumor promoting miRNAs for GBM) in astrocytes co-cultured with the GBM8 neurospheres (data not shown). However, we observed miR-21 transfer when GBM8 neurospheres were co-cultured

with primary astrocytes established from the miR-21 knockout mice. In astrocytes with null miR-21 background, miR-21 became readily detectable upon their co-culture with glioma cells (Fig. 8d). Furthermore, miR-21 transfer led to functional effects, i.e. repression of its previously validated mRNA targets (Fig. 8e) [14]. This data is indicative of the horizontal transfer and suggest that, despite the apparently low copy number of miRNAs per EV, they may exert regulatory functions in the recipient cells, albeit specific conditions and highly sensitive methodologies are required for their detection. The new data is now added in Fig. 8d-e and described on page 13-14 of the revised manuscript.

6. There is little evidence to show that the mRNA targets are validated in Fig. 8. In addition to this, the statement in page 19, “We identified miRNAs with the highest potential for affecting normal brain cells...” would be an overstatement without supportive evidence.

Response:

As described above, our data support the functional miRNA transfer between GSC and normal cultured cells, albeit at a low level which requires very sensitive technologies for detection. However, it is expected that *in vivo*, in the intracranial microenvironment characterized by heterogeneity of cells within the tumor, the close proximity of tumor cells to normal cells, and complex intercellular interactions, different rules may apply to exRNA-mediated communication. While we and others are currently developing the strategies to examine this type of communication in the animal brain, here we predict the miRNAs which transfer from GBM to different types of normal cells would have a major impact, and define them as potentially impactful. The predictions are based on the analysis of differentially expressed miRNAs (in GSC exRNA vs. normal recipient cells), and their previously validated targets. The validation of targets is based on CLIP-Seq datasets available through starBase, one of the most reliable resources for target identification. To be on the conservative side, we only selected as “validated targets” the mRNAs that were found to directly bind miRNAs in at least three CLIP-Seq datasets. We have now revised the text on pages 12 and 20 to better explain this analysis, and modified the text in Discussion to avoid any overstatements.

7. The authors claim that the results of this study will provide a “catalog of diverse vesicular and extravesicular exRNA species useful for biomarker discovery”. In addition to this, they state repeatedly that MVs provide a more accurate read-out of the source cell content than exosomes and could be utilized for identification of biomarkers. This claim could be strengthened by additional evidence that show RNA profiles (or at least a particular read-out) of MVs isolated from patient samples such as blood, cerebrospinal fluid, urine.

Response:

We thank the Reviewer for this great suggestion. We have now tested our filtration method on Cerebrospinal Fluid (CSF) samples obtained from four GBM patients, and included the new data in the revised manuscript. As shown in Fig. 7g, sufficient amounts of total RNA for the multiplex qRT-PCR analysis can be isolated from CSF fractions (MVVs, exosomes, and RNPs), and distinct profiles observed in these fractions for different exRNA classes (mRNA, miRNA, Y RNA, and tRNA). Although we were not able to examine the resemblance of CSF exRNA profiles to those of parental tumor and brain cells due to unavailability of the tumor and brain material from the corresponding

patients, the data indicate that, indeed, MVs and exosomes should be analyzed separately in biomarkers discovery studies. As we speculated in the initial version of our Manuscript, and as we now show in the new Fig. 7g, mRNAs are preferentially associated with MVs, whereas miRNAs with exosomes, and thus MV enrichment is warranted for the mRNA biomarkers, whereas exosomes represent a superior source of miRNA biomarkers. The preference for the full-length Y RNA in MVs and fragment 5'Y RNA in RNPs is also consistent with the observation made on the GSC-derived exRNA. We have added this new data to Fig. 7g and the corresponding text to pages 11-12 and 15 of the manuscript.

8. A major concern is the stability of RNA from MVs, exosomes, and RNP complexes. MVs may present a highly promising source of liquid biopsy biomarkers, but can they still be utilized from stored patient samples? What should be taken into consideration when using patient samples for detecting RNA from MVs over relatively stable miRNA in exosomes?

Response:

As a matter of fact, MV-associated RNA is quite stable. Although we do not have data to directly compare the RNA stability in MVs and exosomes, we have not observed any evidence of reduced RNA stability in MVs relative to exosomes. In the newly added data (Fig. 7g), different fractions were separated from CSF samples stored at -80C for 12-24 months.

We have added more details about our CSF analysis in the Method section, on pages 26-27 of the revised manuscript.

9. Sequential filtration, in addition to the procedures that were used for RNA extraction and library preparation may have caused RNA degradation and fragmentation. How would the authors consider the loss of intact RNA by such methods?

Response:

The procedure of sequential filtration is unlikely to cause RNA fragmentation, since the potential shear force applied is relatively low and insufficient for breaking the vesicles and affecting the RNA; nevertheless, we attempted to investigate this important point experimentally. We now include new data demonstrating that total exRNA isolated from unfractionated conditioned media (cell-free 2,000g-supernatant), not subjected to filtration procedure, exhibits the same level of fragmentation as that observed in filtrated exosomes/RNPs. As shown in Fig. S13, the fragment to full-length ratios were much higher in the unfractionated exRNA than in the cellular RNA, indicating that the fragmentation is a feature of exRNA rather than the artifact of the filtration procedure. In addition, we should note that the most common extracellular RNA fragments are highly specific (e.g. 30nt 5' fragments of Y RNA and tRNA, Fig. 6c and Fig. S12), which also reinforces the idea of selective packaging into exRNA fractions rather than unspecific degradation.

Minor comments:

1. Fig. 2c is not described in the text.

Response:

Sorry for omitting this information. We have now added the following text to the Page 5: "Distinct profiles of several protein markers exhibited by cellular and extracellular fractions further

confirmed the purity of fractions and the lack of cellular contamination in the MV and other extracellular fractions (Fig. 2b).”

2. Overlooked typo in page 5 “The caveat s that this procedure...”

Response:

We have corrected this typo.

3. Inappropriate use of italics used in page 6 “Whereas in 3 out of 4 GSC cultures cellular mRNA reads accounted for...”

Response:

We have corrected this typo.

4. Fig. 4c does not seem to support or agree with Fig. 4b. The length distribution of RNA in MV, exosome and RNP fractions do not seem to differ in Fig. 4c, which is conflicting with Fig. 4b, which shows the different lengths of RNA in the extracellular fractions. The authors should modify the figure to clarify this point.

Response:

There is no contradiction between the data shown in Fig. 4b and 4c. We apologize for the cursory description of the data in the original version of the manuscript that caused the confusion, and now extend the text to clarify this point, as following:

“As shown in Fig. 4b, near-complete short mRNAs could be detected in both cellular and extracellular fractions, but long mRNAs above 1,000 nt were detected in the MVs only. These RT-PCR results suggest that either long mRNAs are excluded from packaging into exosomes and RNP complexes, or they are present in these fractions only as fragments. The latter appears to be the case, since the RNAseq demonstrates a similar representation of transcripts of various lengths in the intracellular compartment and extracellular fractions (Fig. 4c). These data suggest that most exRNA reads corresponding to long mRNAs, in fact, derive from the fragmented extracellular transcripts.”

5. What do the scale bars represent in the heat map cluster analysis in Fig. 6a, 6d and 6f?

Response:

The scale bars in these panels (Fig. 7a, 7d, and 7f of revised Manuscript) represent the percentage of non-rRNA annotated reads, mRNA abundance and log transformed miRNA abundance, respectively. Different scale bars were used here to show the heatmap more clearly. We have updated the figure legend and the related supplemental figures.

6. Fig. 6b requires correction. The angle of the PCA plot is not appropriate.

Response:

We have replotted the PCA in Fig. 7b of revised Manuscript. The corresponding section of the Methods has been revised, as we now used R, instead of the MultiExperiment Viewer, to plot the 3D PCA figure.

7. Fig. 7c is unclear. What do the authors mean in their claim that the mapping of Y1 is “precisely processed”? The figure is unclear, and needs more contextual information.

Response:

Fig. 6c shows the mapping coverage of the Y1 sequencing reads. Higher blue bars indicate increased reads' coverage of the region. As shown in the RNP panel of Fig. 6c, the first 30 nucleotides from the 5' end are highly covered (i.e. produced multiple reads), while the downstream nucleotides are not covered, producing a steep cliff-like coverage curve. This means that all Y1 reads derived from the RNP sample end at the same 3' position, and suggests the presence of specific, precisely processed 5'-Y1 RNA fragments., while a more gradual coverage curve would suggest multiple processing/fragmentation positions. We have now modified the figure and clarified its legend and the corresponding text (page 10).

8. Some terms are vague or not appropriate in the scientific context, and should be replaced with appropriate terms that hold scientific meaning and could be understood in the context of this paper. E.g. page 13 “cell ecosystems”, “indigenous cancer cells”, “transcriptional programs”.

Response:

We proofread the manuscript, revised it, and specified the terms such as those mentioned by the Reviewer.

9. The use of terms “RNA species”, “RNA classes”, “RNA fractions”, should be consistent throughout the text.

Response:

We used the term “RNA class” to define major RNA categories, such as the classes of miRNA, tRNA, Y RNA, etc. The term “RNA species” was used to define individual RNA species, such as miR-21, GluCTC, Y1 RNA, etc. The term “RNA fraction” was used to describe RNA extracted from different extracellular fractions, such as MV RNA, exosome RNA, and RNP RNA. We have now revised the terminology and checked the manuscript for consistent use of the terms.

10. Could apoptotic vesicles also be contained in the MV fraction?

Response:

According to the established literature [15, 16], apoptotic vesicles have a diameter of 1-5 um. If such vesicles existed in the conditioned media of healthy GSC cultures, they should have been removed by filtration through 2.0 um and 0.8 um pores. Since the MV fraction was isolated as a fraction ranging from 0.22 to 0.8 um (Fig. 1a), we believe that it does not contain apoptotic vesicles.

References

1. Wei, Z., et al., *Fetal Bovine Serum RNA Interferes with the Cell Culture derived Extracellular RNA*. Sci Rep, 2016. **6**: p. 31175.
2. Memczak, S., et al., *Identification and Characterization of Circular RNAs As a New Class of Putative Biomarkers in Human Blood*. PLoS One, 2015. **10**(10): p. e0141214.
3. Jeck, W.R., et al., *Circular RNAs are abundant, conserved, and associated with ALU repeats*. RNA, 2013. **19**(2): p. 141-57.

4. Dou, Y., et al., *Circular RNAs are down-regulated in KRAS mutant colon cancer cells and can be transferred to exosomes*. Sci Rep, 2016. **6**: p. 37982.
5. Subramanian, S.L., et al., *Integration of extracellular RNA profiling data using metadata, biomedical ontologies and Linked Data technologies*. J Extracell Vesicles, 2015. **4**: p. 27497.
6. Landgraf, P., et al., *A mammalian microRNA expression atlas based on small RNA library sequencing*. Cell, 2007. **129**(7): p. 1401-14.
7. Consortium, F., et al., *A promoter-level mammalian expression atlas*. Nature, 2014. **507**(7493): p. 462-70.
8. van der Vos, K.E., et al., *Directly visualized glioblastoma-derived extracellular vesicles transfer RNA to microglia/macrophages in the brain*. Neuro Oncol, 2016. **18**(1): p. 58-69.
9. Lai, C.P., et al., *Visualization and tracking of tumour extracellular vesicle delivery and RNA translation using multiplexed reporters*. Nat Commun, 2015. **6**: p. 7029.
10. Quinn, J.F., et al., *Extracellular RNAs: development as biomarkers of human disease*. J Extracell Vesicles, 2015. **4**: p. 27495.
11. Rennert, R.C., F.H. Hochberg, and B.S. Carter, *ExRNA in Biofluids as Biomarkers for Brain Tumors*. Cell Mol Neurobiol, 2016. **36**(3): p. 353-60.
12. Lane, R.E., et al., *Analysis of exosome purification methods using a model liposome system and tunable-resistive pulse sensing*. Sci Rep, 2015. **5**: p. 7639.
13. Cvjetkovic, A., J. Lotvall, and C. Lasser, *The influence of rotor type and centrifugation time on the yield and purity of extracellular vesicles*. J Extracell Vesicles, 2014. **3**.
14. Gabriely, G., et al., *MicroRNA 21 promotes glioma invasion by targeting matrix metalloproteinase regulators*. Mol Cell Biol, 2008. **28**(17): p. 5369-80.
15. Rubartelli, A., A. Poggi, and M.R. Zocchi, *The selective engulfment of apoptotic bodies by dendritic cells is mediated by the alpha(v)beta3 integrin and requires intracellular and extracellular calcium*. Eur J Immunol, 1997. **27**(8): p. 1893-900.
16. Gyorgy, B., et al., *Membrane vesicles, current state-of-the-art: emerging role of extracellular vesicles*. Cell Mol Life Sci, 2011. **68**(16): p. 2667-88.

REVIEWERS' COMMENTS:

Reviewer #1 commented for the editors only and was satisfied by the revision.

Reviewer #2 (Remarks to the Author):

Remarks to the Author

The authors have adequately addressed numbers 2-9 in Major comments. However, a number of issues still remain to be addressed:

The authors used size-based sequential filtration to separate MVs, exosomes, and extracellular RNP complexes. However, one concern is that separating vesicles by size alone may be an oversimplification of categories and the heterogeneous nature of vesicles present in the extracellular environment. Since separation is purely based on particle size, it may be necessary to eliminate the population of other non-EV associated forms. Filtrated compartments could be further separated based on floatation in density gradients at different positions or by immuno-isolation by different surface molecules thereby eliminating the population of other non-EV associated forms.

Could exRNA also be detected from non-EV associated forms such as large protein or lipoprotein complexes? Again, these complexes could also be isolated with EVs by filtration alone.

The addition of new Fig. 1b supports the authors' statement that filtration achieves better separation of EVs and RNPs, and higher yield of RNA over ultracentrifugation. However, in Fig. 1c, what would be the difference between "higher yield" and "higher amount of RNA"?

In Fig. 2b, what are the abbreviations for some of the protein markers identified in the western blot? For instance, what is "La"?

How stable is RNA from RNP complexes and what would be their physiological significance? The authors suggest that there may be selectivity of incorporation of exRNA into MVs, exosomes, and RNPs. It is still unclear why there should be such selectivity, when considering that most of the exRNA is fragmented. A question related to this would be, what could be the significance of enriched YRNA and tRNA in exRNA fractions?

Due to the heterogeneous nature of EV-RNA, one important issue would be to determine which RNA could be used as quality control markers, or house-keeping RNAs are present in each fraction. Is it possible to show any data based on this study?

All minor comments have been adequately addressed by the authors.

REVIEWERS' COMMENTS:

Reviewer #1 commented for the editors only and was satisfied by the revision.

Reviewer #2 (Remarks to the Author):

Remarks to the Author

The authors have adequately addressed numbers 2-9 in Major comments. However, a number of issues still remain to be addressed:

The authors used size-based sequential filtration to separate MVs, exosomes, and extracellular RNP complexes. However, one concern is that separating vesicles by size alone may be an over-simplification of categories and the heterogeneous nature of vesicles present in the extracellular environment. Since separation is purely based on particle size, it may be necessary to eliminate the population of other non-EV associated forms. Filtrated compartments could be further separated based on floatation in density gradients at different positions or by immuno-isolation by different surface molecules thereby eliminating the population of other non-EV associated forms.

We agree that fractionation by sequential filtration cannot provide fully purified EVs and separate their subtypes with 100% efficiency. Nevertheless, as we demonstrate in Figure 1b and 1c, this method is superior to other commonly used methodologies, with exception of density gradient centrifugation. Specifically, 100,000g ultracentrifugation (UC) co-pellets many proteins with EVs, and no exosomal protein marker with exceptional sensitivity and specificity (present on all exosomes but absent on MVs) has been identified in prior studies based on UC. Since the goal of our work was to characterize the exRNA repertoire of principal extracellular complexes (MVs, exosomes, and RNPs), and the amount of exRNA isolated from patient-derived GSCs is very limited, the density gradient centrifugation method (producing further limited RNA amounts per fraction) was impractical for our analysis.

Could exRNA also be detected from non-EV associated forms such as large protein or lipoprotein complexes? Again, these complexes could also be isolated with EVs by filtration alone.

ExRNA associated with large proteins or lipoprotein complexes was mostly isolated in the RNP fraction. Our cut-off of 20nm between exosomes and RNPs corresponds roughly to 500kDa, which is larger than most extracellular proteins or protein complexes. We explicitly acknowledge the limitation of the size-based fractionation on page 4 of the manuscript.

The addition of new Fig. 1b supports the authors' statement that filtration achieves better separation of EVs and RNPs, and higher yield of RNA over ultracentrifugation. However, in Fig. 1c, what would be the difference between "higher yield" and "higher amount of RNA"?

We appreciate this comment and, to be more consistent, have revised the text and removed the "higher amount of RNA" term.

In Fig. 2b, what are the abbreviations for some of the protein markers identified in the western blot? For instance, what is “La”?

We have included full names of proteins in the revised manuscript.

How stable is RNA from RNP complexes and what would be their physiological significance? The authors suggest that there may be selectivity of incorporation of exRNA into MVs, exosomes, and RNPs. It is still unclear why there should be such selectivity, when considering that most of the exRNA is fragmented. A question related to this would be, what could be the significance of enriched YRNA and tRNA in exRNA fractions?

The conditioned media used in this paper was collected from gliomaspheres cultured over 7-10 days, and the profiles exhibited by RNP-associated RNA were consistent and reproducible. Therefore, this RNA fraction is well-protected and stable, at least as much as the EV-enclosed transcripts. Since the presence of the significant amount of exRNA in extravesicular complexes is a new finding emerging from our work and a few other laboratories, its physiological significance is still unclear and certainly warrants further investigations. Regarding the selectivity, the enrichment of specific, rather than random RNA fragments in both EVs and exRNPs, provides strong support for the selectivity. The selectivity of RNA release via MVs, exosomes, or RNPs may root into various intracellular RNA processing pathways and loading mechanisms that should be further elucidated. Clearly, with limited knowledge of the biology of Y RNA and tRNA fragments, the significance of their remarkable extracellular enrichment must be explored beyond this study. This is a fascinating topic; however, to keep this already long manuscript as concise and focused as possible, we refrained from adding this discussion, as not fully irrelevant for this report, but happy to do so per editorial request.

Due to the heterogeneous nature of EV-RNA, one important issue would be to determine which RNA could be used as quality control markers, or house-keeping RNAs are present in each fraction. Is it possible to show any data based on this study?

This is an important question. However, the limited number of biological replicates (four patient-derived neurosphere cultures) and one cell type (GSC), does not allow the identification of the accurate house-keeping RNAs. We have explored several normalization strategies outlined in Laurent LC et al. (PMID 26320937). General recommendation in the field of exRNA is to use spike-in RNAs of different lengths for normalization, along with the house-keeping RNAs that should be determined for a given system (e.g. miR-1246 or U1 snRNA for GSC cellular and extracellular RNA).